# RNase H2, mutated in Aicardi-Goutières syndrome, resolves co-transcriptional R-loops to prevent DNA breaks and inflammation

Agnese Cristini [1], Michael Tellier [1], Flavia Constantinescu[1], Clelia Accalai[1], Laura Oana Albulescu[1], Robin Heiringhoff[1], Nicolas Bery[2], Olivier Sordet[3], Shona Murphy [1] & Natalia Gromak [1 ✉]

RNase H2 is a specialized enzyme that degrades RNA in RNA/DNA hybrids and deficiency of this enzyme causes a severe neuroinflammatory disease, Aicardi Goutières syndrome (AGS). However, the molecular mechanism underlying AGS is still unclear. Here, we show that RNase H2 is associated with a subset of genes, in a transcription-dependent manner where it interacts with RNA Polymerase II. RNase H2 depletion impairs transcription leading to accumulation of R-loops, structures that comprise RNA/DNA hybrids and a displaced DNA strand, mainly associated with short and intronless genes. Importantly, accumulated R-loops are processed by XPG and XPF endonucleases which leads to DNA damage and activation of the immune response, features associated with AGS. Consequently, we uncover a key role for RNase H2 in the transcription of human genes by maintaining R-loop homeostasis. Our results provide insight into the mechanistic contribution of R-loops to AGS pathogenesis.

[1] Sir William Dunn School of Pathology, University of Oxford, South Parks Road, Oxford OX1 3RE, UK. [2] Weatherall Institute of Molecular Medicine, MRC Molecular Haematology Unit, University of Oxford, John Radcliffe Hospital, Oxford OX3 9DS, UK. [3] Cancer Research Center of Toulouse, INSERM, Université de Toulouse, Université Toulouse III Paul Sabatier, CNRS, 31037 Toulouse, France. ✉email: natalia.gromak@path.ox.ac.uk

RNase H1 and H2 are specialized enzymes that degrade the RNA within RNA/DNA hybrids[1]. These hybrids arise as physiological intermediates during DNA replication when RNA is used to prime DNA synthesis of Okazaki fragments. In addition, ribonucleotides (rNMPs) can be mis-incorporated into DNA[2]. Extensive RNA/DNA hybrids are also formed during transcription when the nascent RNA reanneals with the DNA template displacing the single-stranded DNA (ssDNA), forming an R-loop structure[3,4]. Either accumulation of rNMPs in genomic DNA or R-loop formation trigger replication-stress and genomic instability[2–4]. Both RNase H enzymes are essential in mammals; RNase H1 is principally required for mitochondrial DNA replication[5], while RNase H2 is essential for embryonic development[6].

RNase H2 is the main source of RNase H activity in the cell nucleus[7]. Human RNase H2 forms a nuclear heterotrimeric complex composed of the catalytic H2A subunit and two accessory subunits, H2B and H2C, necessary for enzymatic activity[8–11]. Both RNase H2 and H1 can degrade the RNA within the RNA/DNA hybrids[1]. However, only RNase H2 is capable of ribonucleotide excision repair (RER) by removing single rNMP from genomic DNA[2]. Several studies have investigated the biological impact of these two RNase H2 activities but the full picture remains unclear. An RNase H2 mutant, which is RER defective (RED) but still able to process R-loops[12], has revealed that RER activity is required during mouse development[13]. Loss of RER results in accumulation of rNMPs in yeast, murine and human genomic DNA and aberrant processing of rNMPs by topoisomerase I (TOP1), leading to DNA damage and deletions[6,14–19]. However, defects in yeast which lack RNase H2 include loss of heterozygosity, methyl methanesulphonate (MMS) sensitivity, synthetic sickness phenotype induced by combined deletion with the homologous recombination repair helicase Sgs1 and hyper-recombination[12,20–24]. These were all found to be at least partially dependent on defective R-loop resolution. In addition, loss of RNase H2 causes a transcriptional block and R-loop accumulation over rDNA in yeast deficient for Top1 and Rnh1[25] as well as transcriptional down-regulation of genes with R-loop-prone G/C-rich promoters[26].

Despite multiple lines of evidence, a replication-independent function of RNase H2 has not yet been directly investigated. This is important because RNase H2 is mutated in >50% of the cases of Aicardi-Goutières syndrome (AGS), an autosomal recessive inflammatory encephalopathy, which resembles congenital viral infection[27]. AGS is also caused by mutations in other DNA/RNA metabolism enzymes, encoded by *TREX1*, *SAMHD1*, *ADAR1* and *IFIH1*, connecting the accumulation of unprocessed nucleic acids with activation of the AGS-associated immune response[28]. The inflammatory response in *Rnh2* AGS mouse models is dependent on the cGAS-STING pathway[29,30]. Recently, it has been shown that DNA damage primes the induction of the STING-mediated immune response by releasing DNA into the cytoplasm[31,32]. In line with these findings, AGS caused by *RNASEH2* mutations has been mainly attributed to genome instability and a p53-dependent DNA-damage response due to accumulation of rNMPs in genomic DNA[6,14,33,34]. However, transcription itself can be a source of DNA damage, especially in non-replicating cells, where it is involved in neurological diseases[35,36]. Therefore, understanding the function of RNase H2 in nucleic acid metabolism and the role of RNase H2 in transcription and resolution of co-transcriptional R-loops is important to determine whether these processes contribute to AGS.

Here, we show that RNase H2 plays a role in the resolution of co-transcriptional R-loops in human cells. RNase H2 is recruited to a specific subset of active genes in a transcription-dependent manner and interacts with RNA Polymerase II (Pol II). RNase H2 depletion impairs transcription and triggers accumulation of R-loops at specific genomic loci. Notably, R-loops accumulated in RNase H2-deficient cells are processed by structure-specific endonucleases and contribute to activation of the immune response.

In summary, our findings reveal a role for RNase H2 in transcription and provide additional insight into the molecular mechanisms of AGS pathology.

## Results

**RNase H2 binds to active genes genome-wide.** Previous studies have demonstrated that RNAi-mediated depletion of RNase H2 in human cell lines phenocopies RNase H2 dysfunction in AGS patient-derived cells[14]. Therefore, to study RNase H2 function, we have employed siRNA-mediated knock-down that did not affect DNA replication in HeLa cells (Supplementary Fig. 1a, b). As previously described, depletion of each individual RNase H2 subunit resulted in a decrease of both RNase H2A and H2C proteins (Fig. 1a, b and Supplementary Fig. 1c), indicating destabilization of the whole complex[6,14]. Furthermore, both RNase H2A and H2C associated with chromatin (Supplementary Fig. 1d, e) in line with studies in yeast[20,21] and human cells[37]. To investigate its chromatin-associated function, we defined the genome-wide distribution of RNase H2A by chromatin immunoprecipitation (ChIP)-seq in HeLa cells. RNase H2A was enriched in 34,344 genic regions, representing both non-coding and protein-coding genes (Fig. 1c). We further investigated RNase H2A binding over small nuclear (sn) RNA and protein-coding genes, which both have well defined cellular functions. Interestingly, meta-profile of snRNA genes revealed a prominent RNase H2A peak close to transcription start sites (TSS) and to a lesser extent at termination regions (TES) (Fig. 1d, e), supported by ChIP-qPCR on *RNU1* and *RNU2* (Fig. 1f and Supplementary Fig. 1f). Intron-containing protein-coding genes also showed a strong association of RNase H2A with gene promoters and an enrichment over the gene body and termination regions (Fig. 1g and Supplementary Fig. 1g). ChIP-qPCR confirmed RNase H2A binding over the promoters of *ACTB* and *DDX1* (Fig. 1h and Supplementary Fig. 1i). RNase H2A binding was strongly decreased following siRNA-mediated depletion indicating the specificity of the ChIP signal (Fig. 1h). RNase H2 binding was independent of the presence of introns, as meta-analysis highlighted a strong association of RNase H2 with histone and intronless genes (Fig. 1i–l, and Supplementary Fig. 1h). Unexpectedly, we also observed that RNase H2 binding mirrored Pol II occupancy, as determined by ChIP-seq, for all analyzed genes (Fig. 1e and Supplementary Fig. 1g, h). Thus, *TFF1*, which is inactive in HeLa cells, displayed only background levels of RNase H2A binding (Fig. 1h, Supplementary Fig. 1g, i). In the nucleus RNase H2 is found as a fully assembled complex[38]. Consistently, RNase H2C displayed a similar enrichment to RNase H2A by ChIP-qPCR in all gene categories, indicating that RNase H2 binds chromatin as a trimeric complex (Supplementary Fig. 1j–m). Thus, our data revealed that RNase H2 complex is associated with active genes and its correlation with Pol II binding suggests an unexpected role of RNase H2 in transcription in human cells.

**RNase H2 binding to active genes is transcription-dependent.** To investigate the function of RNase H2 in transcription, we further examined the RNase H2 and Pol II binding profiles. Interestingly, RNase H2A binding displayed a significant positive correlation with Pol II occupancy ($r = 0.55$, $p < 0.0001$) and ~85% of RNase H2-associated genes are transcribed by Pol II (Fig. 2a). These observations are consistent with recruitment of RNase H2 to transcribed regions. Blocking transcription reduced Pol II and

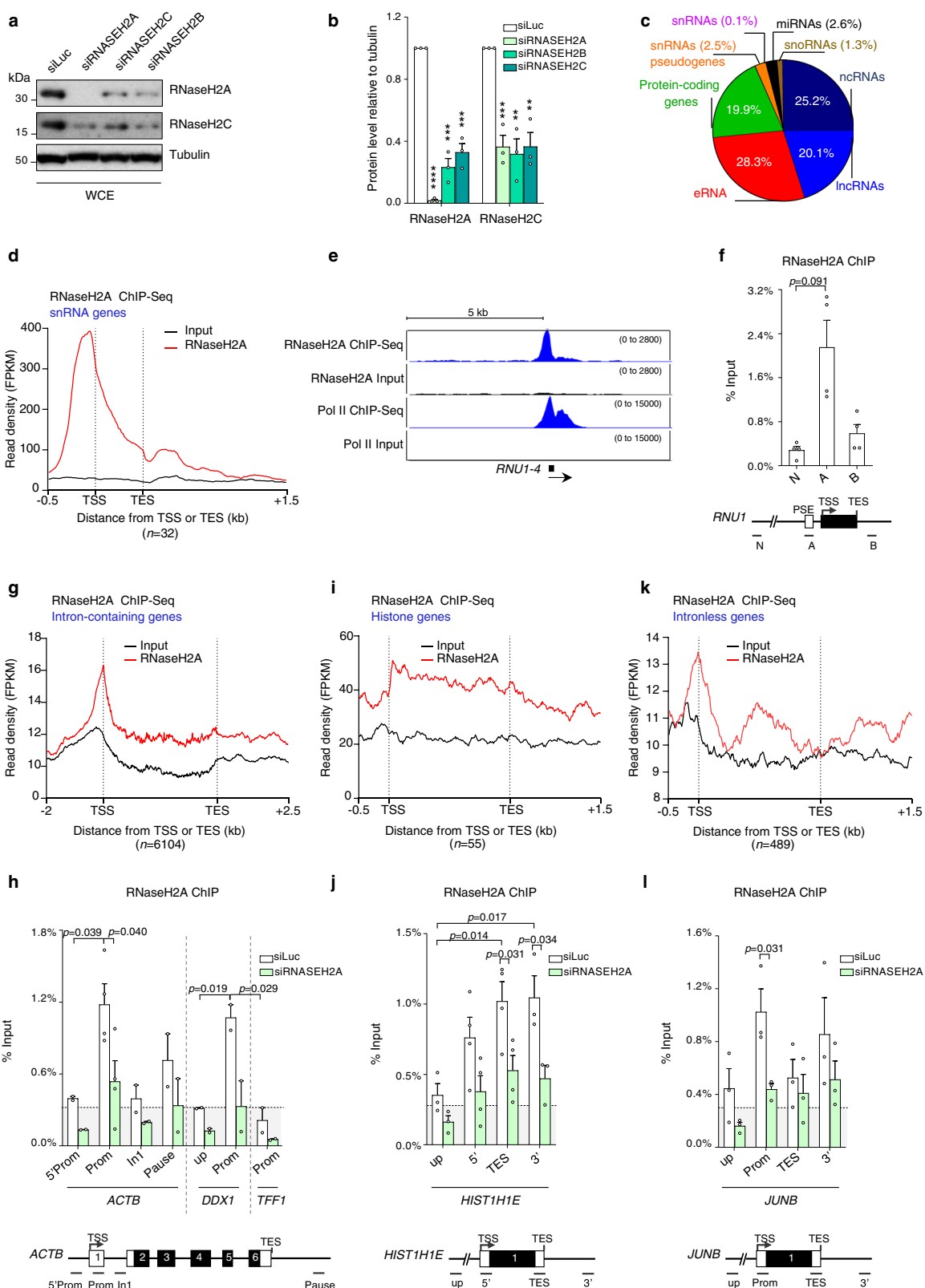

RNase H2A association with *ACTB* and *HIST1H1E*, suggesting that RNase H2 binding at these genes requires active transcription (Fig. 2b–d, g). Next we examined if replication is required for RNase H2 association with transcribed genes. ChIP-qPCR showed that RNase H2A binds *RNU1* and *ACTB* genes in non-replicating fibroblasts, induced into a quiescent state by serum-starvation (Fig. 2e–g). This is in line with previous studies in

yeast, indicating a replication-independent association of RNase H2 with chromatin in G1 phase of the cell cycle[21]. Similarly, RNase H2C binding was detected on *ACTB* and *RNU1* (Supplementary Fig. 2a). These findings indicate that binding of the RNase H2 complex to active genes does not require replication. Indeed, HeLa cells treated with low doses of the DNA Pol inhibitor aphidicolin (APH), that blocks DNA replication

**Fig. 1 RNase H2A binds active genes. a, b** Western blot of RNase H2A and H2C in the whole cell extract (WCE) in HeLa cells transfected with siLuc (white), siRNASEH2A, H2B and H2C (shades of green). Tubulin is a loading control. **b** Quantification of protein levels. Values are normalized to tubulin. $n = 3$ biologically independent experiments (means ± SEM). $p$-values from left to right: $p < 0.0001$, $p = 0.0002$, $p = 0.0003$, $p = 0.001$, $p = 0.002$, $p = 0.002$ (two-tailed unpaired $t$-test). **c** Distribution of RNase H2A ChIP-seq positive genes across the indicated genic compartments ($n = 34,244$). **d** Meta-analysis of read density (fragments per kilobase of transcript per million mapped reads, FPKM) for RNase H2A ChIP-seq on snRNA genes across −0.5 kb genomic region flanking transcription start site (TSS) and +1.5 kb transcription termination site (TES) in HeLa cells. All subsequent images show one representative replicate out of 2 with RNase H2A signal in red and input in black. Number of genes analyzed are shown in brackets. **e** RNase H2A and Pol II ChIP-seq profiles of *RNU1-4* in HeLa cells. Numbers in brackets indicate the viewing range (FPKM). **f** RNase H2A ChIP-qPCR for *RNU1* in HeLa cells. Values are expressed as percentage of input. $n = 4$ biologically independent experiments (means ± SEM). $p$-values were calculated using two-tailed unpaired $t$-test. All subsequent gene diagrams depict snRNA coding region or protein-coding gene exons (black), UTRs (white), introns (lines), TSS, TES and qPCR amplicons (below the diagram). PSE is proximal sequence element. **g, i, k** Meta-analysis of read density (FPKM) for RNase H2A ChIP-seq on different categories of genes in HeLa cells, with the window size of the presented plots adjusted to the gene size. Data are represented as in **d. h, j, l** RNase H2A ChIP-qPCR in HeLa cells transfected with siLuc (white bars) and siRNASEH2A (green bars) for different gene categories. Values represent percentage of input (means ± SEM). $p$-values were calculated using two-tailed unpaired $t$-test. **h** Intron-containing genes (*ACTB, DDX1* and *TFF1*). $n = 2$ ($n = 4$ for *ACTB* Prom) biologically independent experiments. **j** Histone *HIST1H1E* gene. $n = 3$ ($n = 4$ for 5′ and TES) biologically independent experiments. **l** Intronless *JUNB* gene. $n = 3$ biologically independent experiments. Horizontal dotted line indicates background signal. Source data are provided as a Source data file.

(Supplementary Fig. 2b–d), still maintained RNase H2A binding over *ACTB* and *HIST1H1E* (Supplementary Fig. 2b, e). Interestingly, a small decrease in RNase H2A binding following APH treatment correlated with a decrease in Pol II levels, again suggesting a connection with transcription (Supplementary Fig. 2f). In agreement with these observations, both Actinomycin D and the Pol II inhibitor α-Amanitin strongly reduced RNase H2A and Pol II binding to the *RNU1* and *ACTB* in non-replicating fibroblasts (Fig. 2b, e–g, Supplementary Fig. 2b, g, h). Taken together, our results suggest that RNase H2 may play a role in gene transcription.

**RNase H2 interacts with Pol II and enhances transcription**. To examine the biochemical connection between RNase H2 binding and transcription, we carried out co-immunoprecipitation (co-IP) of endogenous RNase H2A in HeLa cell extracts treated with benzonase, which degrades all nucleic acids, but still retains protein-mediated interactions. As expected, RNase H2C interacted with RNase H2A (Fig. 3a). RNase H2A co-IP also contained Pol II as detected by four independent antibodies, which recognize un-phosphorylated or transcription-associated Ser2/Ser5-phosphorylated Pol II isoforms (Fig. 3a). Co-IP of endogenous RNase H2C confirmed these results (Supplementary Fig. 3a). Interestingly, Pol II interacted with RNase H2A also in non-replicating fibroblasts (Supplementary Fig. 3b), indicating that this interaction does not require replication. We have also observed an interaction of RNase H2 with Pol I, suggesting that RNase H2 may be involved in rDNA gene transcription (Supplementary Fig. 3c).

To determine whether RNase H2 affects transcription, we quantified nascent RNA synthesis using 5-ethynyl uridine (EU) labeling. This analysis revealed a significant decrease of global transcription in RNase H2A-depleted HeLa and HEK293T cells (Fig. 3b, c and Supplementary Fig. 3d, e). This was also observed for the nuclear EU signal outside of nucleoli, which mainly reflects nascent Pol II transcripts[39] (Fig. 3c and Supplementary Fig. 3e; middle panel), and nucleolar EU signal, associated with rDNA transcription (Fig. 3c and Supplementary Fig. 3e; right panel). These observations are consistent with RNase H2 interacting with both Pol II and Pol I polymerases. Notably, Pol II binding to chromatin was also reduced in RNase H2A-deficient cells (Supplementary Fig. 3f, g). To further confirm these results, we carried out chromatin-bound RNA (chrRNA)-seq, which mainly enriches for nascent transcripts genome-wide[40]. In line with our EU data, the chrRNA-seq signal was significantly decreased ($p < 0.0001$) in RNase H2-deficient cells for all gene categories, especially snRNA, histone and intronless genes as based

on both meta-profile and individual gene analysis (Fig. 3d, f–h). The effect was less pronounced for intron-containing genes, where the most (~90%) showed <1.5-fold signal change (Fig. 3e, h). Taken together, our results reveal that RNase H2 associates with the Pol II complex which acts to promote nascent RNA transcription.

**RNase H2 promotes co-transcriptional R-loop resolution**. Several studies have demonstrated that RNA/DNA hybrids can impair Pol I and Pol II transcription in vitro[41], in bacteria[42], in yeast[25,43] and in human cells[44,45]. Therefore, impaired Pol II transcription in RNase H2-deficient cells may be associated with defective RNA/DNA hybrid processing. To examine this, we employed RNA/DNA hybrid immunoprecipitation (DRIP)-seq with the S9.6 antibody, which recognizes RNA/DNA hybrids[44,46]. We detected 56,997 R-loop peaks in siLuc and 50,955 peaks in siRNASEH2A cells, mapping primarily in genic regions, including protein-coding genes, long non-coding (lnc) RNAs and enhancer (e) RNAs (Fig. 4a and Supplementary Fig. 4a), consistent with previous studies[40,46]. Interestingly, meta-analysis revealed a strong enrichment of RNase H2A occupancy over DRIP peaks (Fig. 4b). This observation is confirmed by comparative RNase H2A ChIP-seq and DRIP-seq occupancy on multiple genes (Supplementary Fig. 4b, c), suggesting that RNase H2 binds to R-loop-enriched regions. Furthermore, we identified 16,647 'increased/gained' and 35,604 'decreased/lost' DRIP peaks in RNase H2-depleted cells. The majority of changes occurred within the genic regions, with 81% of genic DRIP peaks, associated with 21,509 unique loci, being affected, in line with a role for RNase H2 in transcription (Fig. 4a, Supplementary Fig. 4a). Since changes in R-loop load influenced primarily protein-coding genes (Fig. 4a) and both snRNA and protein-coding genes displayed RNase H2A binding (Fig. 1), we examined these genes in more detail. Meta-gene analysis revealed an increased R-loop signal over snRNA genes in RNase H2A-deficient cells as confirmed by DRIP-qPCR in cells depleted for individual RNase H2 subunits (Fig. 4c–e). The DRIP-seq signal was RNase H-sensitive confirming its specificity (Fig. 4c, d). Following RNase H2A depletion, R-loop signal increased in all protein-coding gene categories (Supplementary Fig. 5a). For example, R-loop accumulation was observed on *HIST1H1E, HIST1H2BG* and intronless *JUNB* (Fig. 4f, Supplementary Fig. 5b–e). Intron-containing genes showed a more subtle change in R-loop levels with only ~30% of genes displaying >1.5-fold increase in the signal (Supplementary Fig. 5f, g). Intriguingly, RNase H2-depleted cells preferentially accumulated R-loops in short (<2 kb) intron-containing genes (Supplementary Fig. 5h, i). *ACTB* and *HNRNPL*, both longer than 2 kb, showed unchanged R-loop

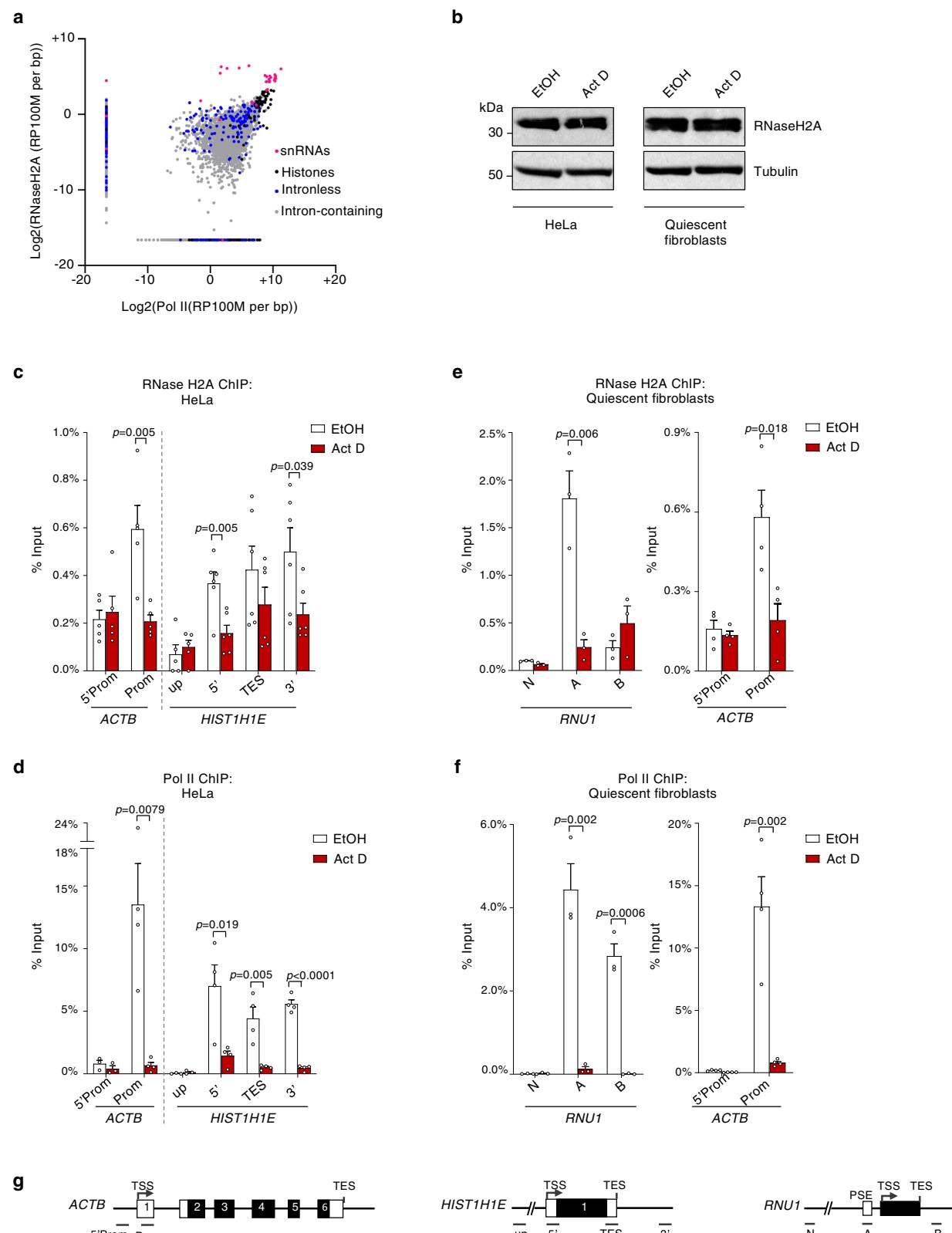

levels upon RNase H2 depletion (Fig. 4f right panel and Supplementary Fig. 5g). This suggests that despite RNase H2 binding to these genes, either its R-loop-resolving activity is not necessary or it can be compensated by other factors. Consistent with short genes being affected stronger by RNase H2 depletion, protein-coding genes positive for RNase H2A binding in ChIP-seq are significantly shorter compared to RNase H2A-negative genes

(Supplementary Fig. 5j). Interestingly, RNase H1 over-expression in RNase H2-depleted HEK293T cells led to decreased R-loop levels, further supporting the role of RNase H2 in promoting co-transcriptional R-loop resolution (Fig. 4g, h and Supplementary Fig. 6a, b). This RNase H1 over-expression did not affect the recruitment of RNase H2 to these genes (Supplementary Fig. 6c, d). Our data indicate that RNase H2 is required to prevent

**Fig. 2 RNase H2A binding is transcription-dependent. a** Scatterplot of RNase H2A versus Pol II ChIP signal across genes, defined as TSS to TES. Genes with RNase H2A or Pol II ChIP signal positive over the input are shown. snRNA (pink), histone (black), intronless (blue) and intron-containing (gray) genes are shown. **b** Western blot of RNase H2A in HeLa cells and quiescent fibroblasts treated with actinomycin D (Act D) and ethanol (vehicle). Tubulin is a loading control. Representative blots from $n = 2$ biologically independent experiments. **c** RNase H2A ChIP-qPCR and **d** Pol II ChIP-qPCR of *ACTB* and *HIST1H1E* genes in HeLa cells, treated with ethanol (vehicle; white bars) or actinomycin D (Act D; red bars). Values represent percentage of input (means ± SEM). *p*-values were calculated using two-tailed unpaired *t*-test. $n = 5$ or 6 biologically independent experiments in **c**. $n = 3$ or 4 biologically independent experiments in **d**. **e** RNase H2A ChIP-qPCR and **f** Pol II ChIP-qPCR of *RNU1* and *ACTB* genes in WI38 hTERT cells (quiescent fibroblasts), treated with ethanol (vehicle; white bars) or actinomycin D (Act D; red bars). Values represent percentage of input (means ± SEM). *p*-values were calculated using two-tailed unpaired *t*-test. $n = 3$ (*RNU1*) or $n = 4$ (*ACTB*) biologically independent experiments. **g** Diagrams of the indicated genes. Source data are provided as a Source data file.

R-loop accumulation at a specific subset of genes; primarily short and intronless genes.

**R-loops contribute to the inflammatory response.** Accumulation of endogenous nucleic acids in the cytoplasm triggers immune activation characteristic of AGS[28]. Recently, DNA damage was identified as an important source of immunogenic nucleic acids[31,32,47,48]. Since RNase H2 deficiency results in R-loop accumulation and R-loops are a source of DNA damage, we asked whether R-loops could contribute to activation of an immune response by increasing genome instability. Depletion of RNase H2 subunits in HeLa cells significantly induced a panel of inflammation and immune-related mRNAs (*IFNGR1*, *OAS1*, *PTGS2*, *TNF*, *ISG20*, *STING*), previously analyzed in other reports[29,48], without affecting the expression of the control housekeeping genes (Fig. 5a, b, Supplementary Fig. 7a, b). Consistently, we have also observed an accumulation of micronuclei in these cells in line with previous reports[14,49,50] (Supplementary Fig. 7c).

To assess the contribution of R-loops in the activation of immune genes, we over-expressed RNase H1 in RNase H2-deficient HEK293T cells (experimental set-up Fig. 4g). This resulted in decreased R-loops (Fig. 4h and Supplementary Fig. 6b) and diminished the expression of immune-related genes (Fig. 5c). Inversely, depletion of the R-loop resolving helicase AQR[51] induced the expression of immune-related genes, that was further exacerbated by co-depletion of RNase H2A (Supplementary Fig. 7d, e). These data suggest that R-loops contribute to activation of the immune response.

Deficiency of RNase H2 causes accumulation of DNA damage[6,14,33,52]. Consistently, we found that RNase H2 depletion significantly increased DNA double-strand breaks (DSB), measured by neutral comet assay (Supplementary Fig. 8a). We and others have shown that both single-strand breaks (SSBs) and DSBs can arise from the processing of unscheduled and persistent co-transcriptional R-loops by the structure-specific endonucleases XPG and XPF[51,53]. Therefore, to test the contribution of R-loops to the appearance of DNA breaks in RNase H2-deficient cells, we employed alkaline comet assay, which detects both SSBs and DSBs. RNase H2A depletion resulted in increased alkaline comet tail moment which was partially rescued by RNase H1 over-expression, suggesting that R-loops contribute to DNA damage (experimental set-up Fig. 4g; Fig. 5d). Moreover, depletion of XPG or XPF decreased the DNA breaks observed in RNase H2A-deficient cells, indicating that DNA breaks may result from XPG and/or XPF-mediated R-loop cleavage (Fig. 5e and Supplementary Fig. 8b, c). Interestingly, depletion of XPG or XPF diminished the induction of immune-related genes in RNase H2A-depleted cells (Fig. 5f and Supplementary Fig. 8d–g). These findings suggest that R-loops and endonuclease-mediated R-loop cleavage contribute to DNA breaks which activate the immune response in RNase H2-deficient cells, highlighting the potential role of R-loops in AGS pathogenesis.

## Discussion

AGS is a severe autoimmune disease with dramatic effects on the brain caused by the accumulation of immunogenic nucleic acids[27,28]. Understanding the cellular functions of RNase H2, most frequently mutated in AGS, is important to decipher the molecular mechanisms underlying AGS pathology. Our findings reveal a pivotal role of RNase H2 in transcription and R-loop resolution (Fig. 6). The RNase H2 complex binds a specific subset of actively-transcribed genes likely as part of the Pol II complex and maintains the R-loop homeostasis co-transcriptionally, allowing efficient gene expression (Fig. 6a). In the absence of RNase H2, co-transcriptional R-loops accumulate above the physiological level, resulting in transcriptional downregulation and DNA damage, mediated by structure-specific endonucleases, thus activating inflammatory response (Fig. 6b).

Previous studies have shown RNase H2 association with chromatin at telomeres and DNA damage sites[54,55]. However, to date, the genome-wide distribution of RNase H2 has not been examined. Our study represents the first analysis of RNase H2 binding genome-wide in human cells, demonstrating its correlation and dependence on active transcription. We further show that replication is not required for RNase H2 binding, in line with high RNase H2 expression in post-mitotic mice neurons[56] and RNase H2 association with chromatin in G1 in yeast[21]. Consistent with a transcription-associated function of RNase H2, we identified a subset of genes that respond to RNase H2 loss by accumulating R-loops. These observations are in agreement with several reports suggesting a connection between genome instability caused by RNase H2 depletion and R-loops[12,20–25,52]. R-loops are formed during transcription and they are enriched over promoter and terminator regions of human genes[44,46]. However, if they exceed the physiological level, they need to be removed co-transcriptionally to avoid replication stress and transcriptional impairment. Through its association with RNA Pol II, RNase H2 may play a role in transcriptional elongation checkpoint control by ensuring the removal of RNA/DNA hybrids that are harmful for both genome integrity and transcriptional efficiency. Consequently, RNase H2 deficiency may cause a defect in checkpoint control and the accumulation of abortive transcripts forming R-loops, resulting in nascent transcriptional elongation impairment. Persistent R-loops may also reduce the speed of Pol II elongation leading to polymerase pileups/stalling. This is reminiscent of the proposed model for RNase H 1/2 function in transcription of rDNA genes in yeast[25,57]. Indeed, Top1-deficient yeast strains, lacking RNase H activity, accumulate R-loops at rDNA genes. This results in impaired transcriptional elongation, Pol I pileups and reduced rates of pre-rRNA synthesis[25,57]. In addition to the well-described function of RNase H2 in cleaving RNA in RNA/DNA hybrids[1], nuclease-independent functions of RNase H2 cannot be completely excluded. Protein-protein interactions with components of the transcriptional complex may also be important for its role in gene expression. Interestingly, many of the genes accumulating

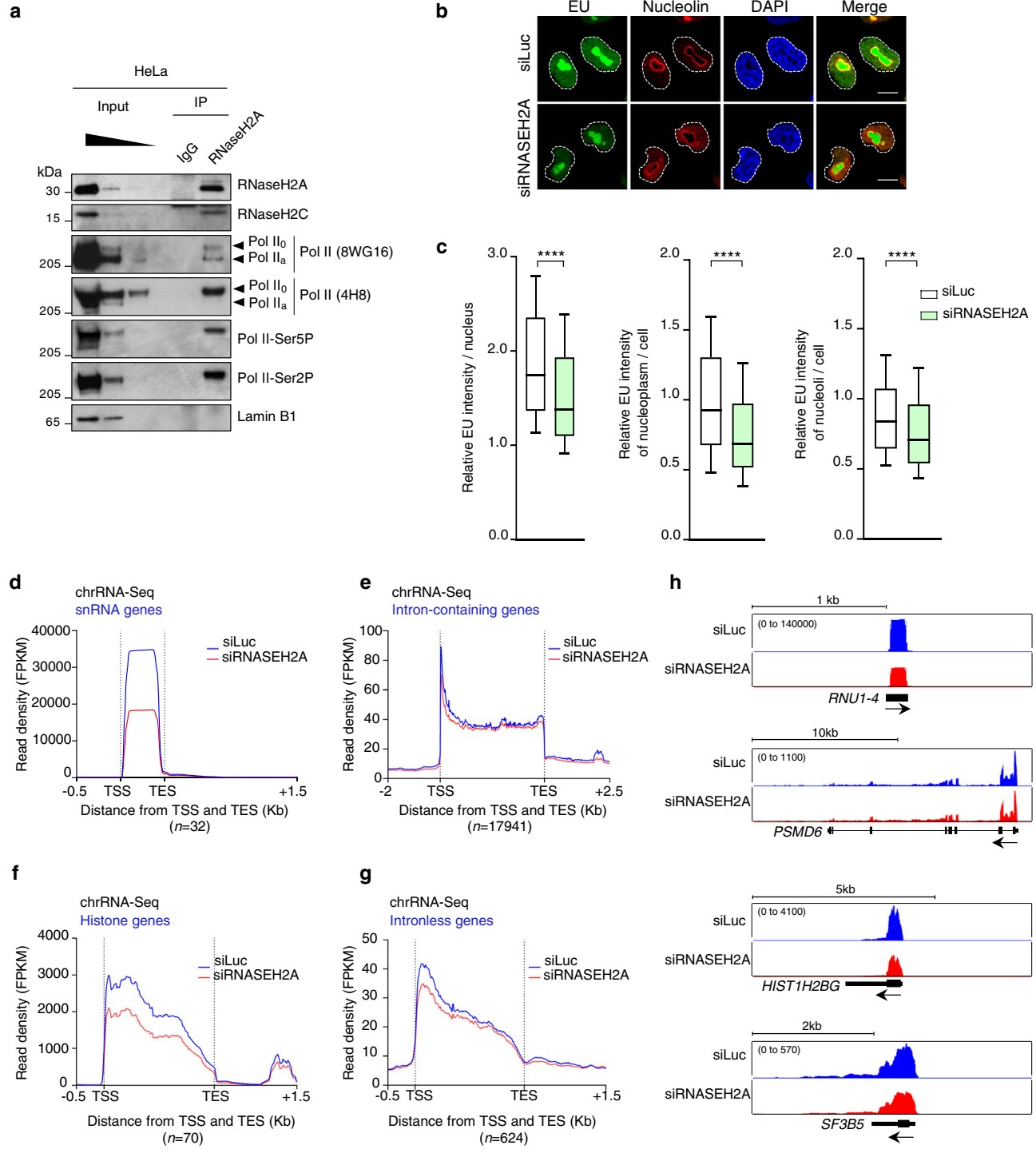

**Fig. 3 RNase H2A depletion affects transcription. a** Western blot of endogenous RNase H2A IP from HeLa WCE probed with the indicated antibodies. Arrows indicate hypo- (IIa) and hyper-phosphorylated (IIo) Pol II. IgG and Lamin B1 are negative controls. Representative blots from $n = 2$ biologically independent experiments. **b** IF analysis of EU incorporation in HeLa cells transfected with siLuc or siRNASEH2A. EU (green), nucleolin (red), DAPI (blue), Nucleolin + EU (merge); scale bars: 10 μm. Representative images from $n = 2$ biologically independent experiments. **c** Quantification of EU intensity per nucleus (left panel), nucleoplasm/cell (nucleus without nucleolus; middle panel) or nucleoli/cell (right panel) in HeLa cells transfected with siLuc (white) or siRNASEH2A (green). EU intensity is normalized to the average nucleoplasmic intensity in siLuc condition. Boxplot settings are: box: 25–75 percentile range; whiskers: 10–90 percentile range; horizontal bars: median; outliers not displayed. >20,000 nuclei were quantified per condition. Representative data from $n = 3$ biologically independent experiments. ****$p < 0.0001$ (two-tailed unpaired *t*-test). **d**–**g** Meta-analysis for chrRNA-Seq on indicated categories of genes in HeLa cells following RNase H2A depletion with the window size of the presented plots adjusted to the gene size. siLuc (blue) and siRNASEH2A (red) tracks show the average signal of two replicates. **h** ChrRNA-seq profiles of *RNU1-4*, *PMSD6*, *HIST1H2BG* and *SF3B5* genes in HeLa cells treated as in **d**–**g**. Source data are provided as a Source data file.

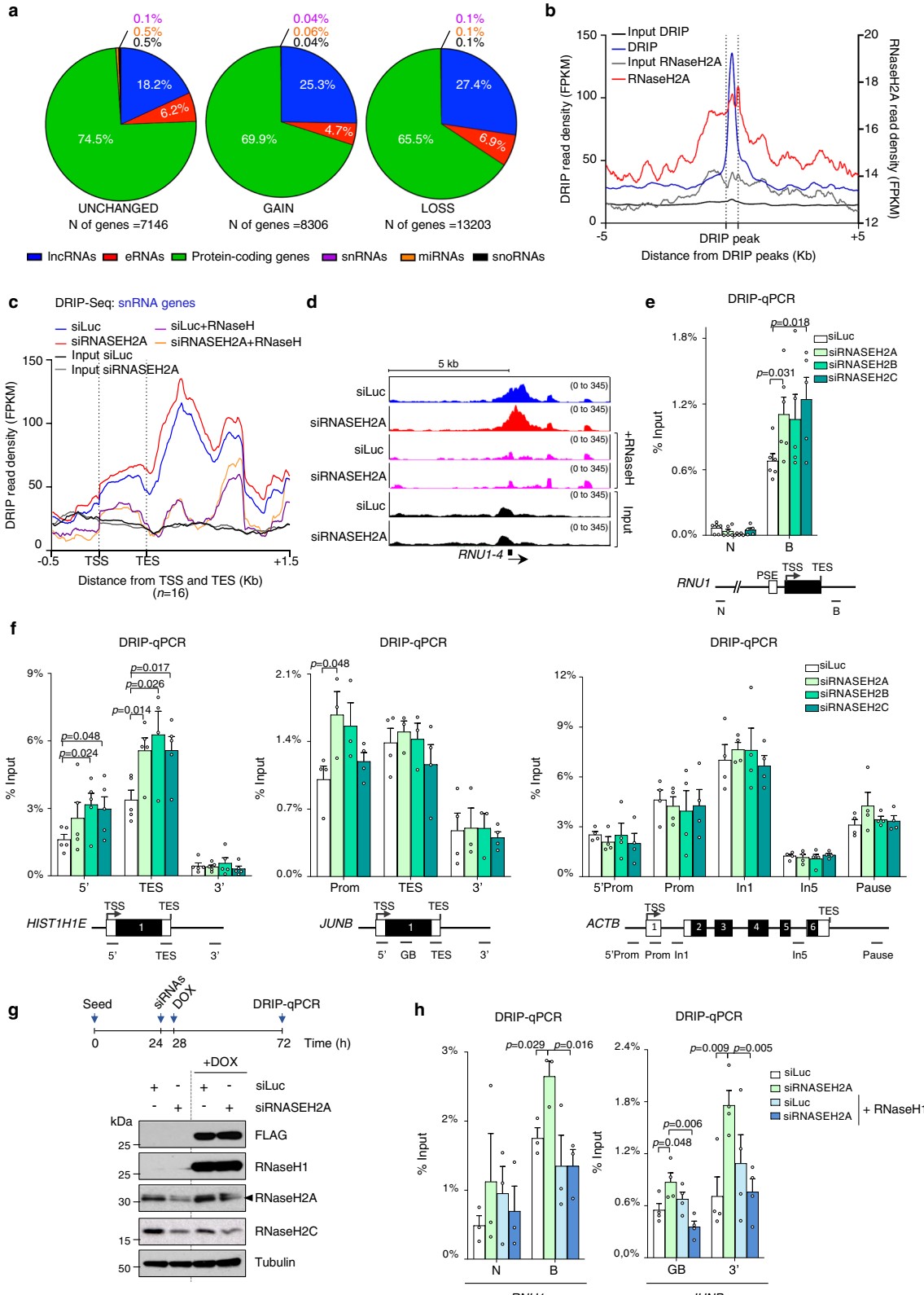

R-loops in RNase H2-deficient cells are short, intronless and highly expressed, such as snRNA and histone genes. This bias towards short genes may be explained by the ability of introns to limit R-loop accumulation through splicing, especially upon deficiency of physiological R-loop regulators, such as THO factors[58,59]. Considering the multitude of R-loop regulators in human cells[60], intron-containing and long genes may employ different and redundant factors to regulate R-loop homeostasis in contrast to short and intronless genes. Indeed, TOP1 efficiently prevents accumulation of co-transcriptional R-loops preferentially in highly transcribed long genes[61]. RNA/DNA unwinding by helicases may also be a preferred way to resolve co-transcriptional R-loops in intron-containing genes compared to the degradation of the nascent RNA. Longer and highly

**Fig. 4 RNase H2 depletion induces R-loop accumulation. a** Distribution of genes in the indicated genic compartments defined by DRIP-seq peaks in control and RNase H2A-depleted HeLa cells. Unchanged (left), increased/gained (middle) and decreased/lost (right) peaks in RNase H2A-depleted HeLa cells are shown. **b** Meta-analysis of DRIP-seq peaks (blue) enriched over both input and RNase H-treated control in siLuc samples (+/−250 bp around peak center). RNase H2A ChIP-seq data (red) are superimposed on DRIP peaks (+/−5 kb). **c** Meta-analysis of DRIP-seq on snRNA genes following RNase H2A depletion in HeLa cells with the window size of the presented plot adjusted to the gene size. siLuc (blue) and siRNASEH2A (red) tracks show an average signal of two replicates. RNase H-treated samples correspond to one replicate and are shown in orange and purple. **d** DRIP-seq profiles of *RNU1-4* in HeLa cells treated as in **c**. **e**, **f** DRIP-qPCR for the indicated genes in HeLa cells following RNase H2A, H2B or H2C depletion (shades of green). Values are expressed as percentage of input (means ± SEM). *p*-values were calculated using two-tailed unpaired *t*-test. **e** snRNA *RNU1* gene. n = 5 (siRNASEH2B and siRNASEH2C) and n = 6 (siLuc and siRNASEH2A) biologically independent experiments. **f** Histone *HIST1H1E* gene; n = 5 (n = 4 TES siRNASEH2B) biologically independent experiments (left panel). Intronless *JUNB* gene; n = 3 (siRNASEH2A and siRNASEH2B) and n = 4 (siLuc and siRNASEH2C) biologically independent experiments (middle panel). Intron-containing *ACTB* genes; n = 4 biologically independent experiments (right panel). Gene diagrams are shown on the bottom panel. **g**, **h** HEK293T RNase H1-FLAG-IRES-mCherry cells transfected with siLuc (white bars) or siRNASEH2A (green bars) and induced with doxycycline (+DOX) to over-express RNase H1 (shades of blue). **g** Schematic of the protocol (top panel), Western blot probed with the indicated antibodies (bottom panel). **h** DRIP-qPCR for the indicated genes. Values are expressed as percentage of input (means ± SEM); n = 3 (*RNU1*) and n = 4 (*JUNB*) biologically independent experiments. *p*-values were calculated using two-tailed unpaired *t*-test. Source data are provided as a Source data file.

transcribed genes are protected from R-loop accumulation by UAP56/DDX39B[62]. SETX, DHX9 and DDX23 helicases are all associated with transcription and R-loop resolution in *ACTB* gene[44,60,63], which did not accumulate R-loops upon RNase H2 depletion. Similarly, other R-loop regulators, including DDX5, XRN2 and PRMT5 share many common R-loops substrates and simultaneously they can also target unique R-loop regions[64], even though their specificity is not dictated by the gene size. Therefore, the presence of redundant R-loop regulators in human cells may underlie the fact that RNase H2-driven R-loop overload was observed only in a specific subset of genes. However, more substantial and widespread R-loop accumulation may exist at early time points after RNase H2 knockdown. Notably, we analyzed R-loops following RNase H2 depletion for 72 h. At this point R-loop removal backup pathways may have already taken place in some genomic regions. Accumulation of R-loops observed in RNase H2-depleted cells implies that endogenous RNase H1 is unable to compensate for loss of RNase H2 activity. Studies in yeast suggested that RNase H1 is involved in the resolution of a small subset of genomic R-loops, including those induced by stress[20,21] and that some R-loops specifically require RNase H2[12]. Notably the ectopic over-expression of RNase H1, which provided physiological levels of RNase H activity, was unable to rescue DNA damage and inflammation phenotypes in murine cells lacking *Rnaseh2b*[30]. Taken together, these and our findings suggest independent functions for the two RNase H enzymes. In line with our results, high level of RNase H1 overexpression may be required to compensate for loss of RNase H2 activity. Further studies to identify genomic loci associated with RNase H1 will help shed light on the differential specificities of RNase H2 versus RNase H1 in human cells.

Accumulation of rNMPs in genomic DNA together with high levels of DNA damage and increased micronuclei formation have all been described as sources of cytoplasmic nucleic acids that activate the cGAS/STING-dependent immune response[6,14,33,49,50]. In contrast, a contribution of endogenous retro-elements to RNase H2-driven AGS phenotype has recently been called into question[50,65]. Our findings add R-loops to the possible sources of immunogenic nucleic acids that may trigger DNA damage and inflammation in AGS. Thus, we show here that persistent R-loops accumulate in the RNase H2-deficient cells and they are converted into DNA breaks through XPF- and XPG-mediated cleavage. Signaling of the DNA damage itself is a key driver of AGS neuropathology, as it was recently demonstrated in a murine model of neural *Rnaseh2b* inactivation[34]. Moreover, DNA damage also triggers an inflammatory response by activating the cGAS–STING pathway, as reported in multiple studies[31,32,47,48]. R-loop

processing could directly release immunogenic ssDNA or RNA/DNA hybrids that are then sensed by cGAS-STING or TLR9 receptors in the cytoplasm[47,66]. Similarly, ssDNA fragments released from stalled replication forks can activate cGAS-STING-dependent inflammatory response in cells deficient for *SAMHD1*, another gene mutated in AGS[48]. It cannot be excluded that persistent R-loops are sensed by cGAS directly in the nucleus, as suggested during development[67]. Harmful R-loops may also lead to an immuno-stimulatory response by favoring micronuclei generation[68] or replication fork stalling[3]. Finally, an intriguing possibility is that rNMPs trapped within the genomic DNA may promote R-loop accumulation in RNase H2-deficient cells as a result of replication stress[14] or TOP1-mediated removal of rNMPs[15–19]. Indeed, this effect could increase the trapping of TOP1 on chromatin and so further induce R-loops[53]. In conclusion, our findings uncover a role for RNase H2 in gene expression that is associated with the maintenance of R-loop homeostasis in human cells. By defining the function of RNase H2 in R-loop metabolism, our work provides insight into how R-loop balance is maintained at physiological levels in human cells and how its dysregulation may contribute to AGS pathogenesis.

## Methods

**Cell culture and treatments**. HeLa and HEK293T cells were obtained from ATCC and grown in DMEM medium supplemented with 10% (v/v) fetal bovine serum (FBS), 1% (v/v) penicillin/streptomycin at 37 °C in 5% CO$_2$. Primary human lung embryonic WI38 fibroblasts immortalized with hTERT were obtained from Carl Mann (CEA, Gif-sur-Yvette, France)[69]. Cells were cultured in modified Eagle's medium (MEM) supplemented with 10% (v/v) FBS, 1 mM sodium pyruvate, 2 mM glutamine, 0.1 mM non-essential amino acids and 1% (v/v) Amphotericin B/penicillin/streptomycin (Thermo-Fisher, 15240-062) at 37 °C in 5% CO$_2$. Quiescence was induced as described[70] by washing the cells with the serum-free medium twice and culturing them in the growth medium (as above) supplemented with 0.2% (v/v) FBS for 72 h. HEK293T-Tet-inducible RNase H1-FLAG-IRES-mCherry cells were generated by transducing HEK293T with the pLVX-TetOne RNase H1-FLAG-IRES-mCherry lentivirus. The plasmid encoding for human RNase H1-FLAG[45] was cloned by replacing the GFP tag in the RNase H1-GFP plasmid provided by Prof. R.J. Crouch. RNase H1-FLAG construct lacks the 26 amino acid corresponding to the mitochondrial localization signal (MLS)[5]. RNase H1-FLAG[45] and IRES-mCherry were cloned in the pLVX-TetOne lentivector (Takara Bio) by PCR using EcoRI/BamHI sites. To induce RNase H1-FLAG and mCherry expression, 1 µg/ml doxycycline was added to the culture medium supplemented with tetracycline-free FBS (Takara Bio, 631102) for 48 h. Drugs and chemicals used are: Act D (Sigma, A9415; 5 µg/ml in HeLa and 2.5 µg/ml in WI38 hTERT cells, 6 h), APH (Sigma, A0781; 1 µM, 1 h), α-aman (Sigma, A2263; 2.5 µg/ml, 24 h), BrdU (Sigma, B5002) and DOX (MP Biomedicals, 198955). Act D was dissolved in ethanol, APH in DMSO and α-aman, BrdU and DOX in water. Control samples were treated with the vehicle only.

**siRNA transfections**. siRNA transfections were performed for 72 h (HeLa) or 48 h (HEK293T) using Lipofectamine 2000 (Thermo Fisher) according to the manufacturer's instructions. HeLa cell transfections were performed in 6-well plates 24 h

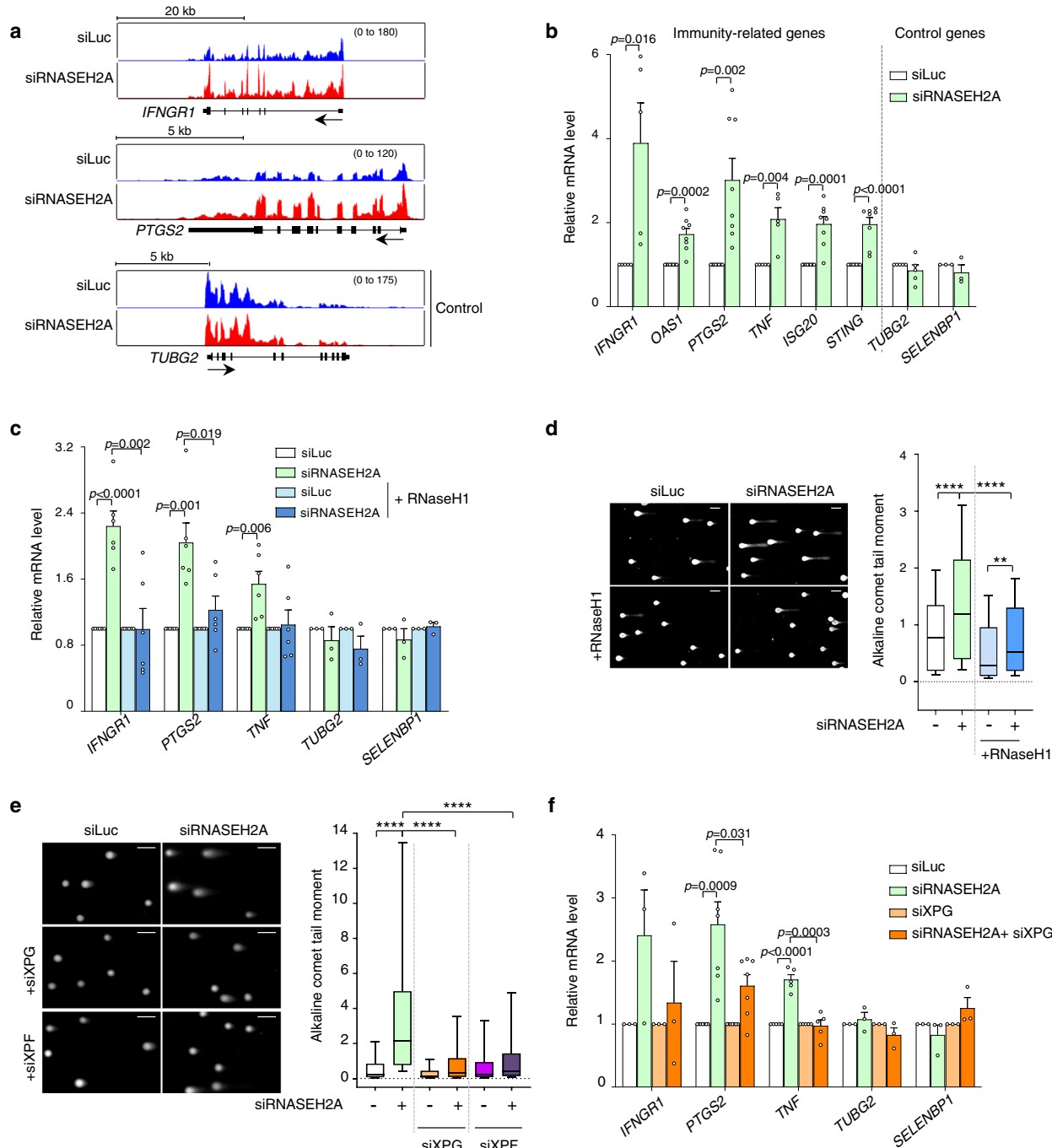

after seeding with 22 nM siRNA and the medium was exchanged 6 h after transfection. 24 h later, transfection was repeated and the cells were re-seeded 24 h before the experiment. For HEK293T cells one round of transfection was performed with 33 nM siRNA. siRNAs were purchased from GE Healthcare targeting firefly luciferase (D-001400-01), *RNASEH2A* (D-003535-01), *RNASEH2B* (AUCAAACUGUGGCAGCAUUAAdTdT), *RNASEH2C* (D-014801-01), *AQR* (D-022214-03), *XPF* (D-019946-02 in Supplementary Fig. 8e–g or M-019946-00 in Fig. 5e) and *XPG* (GAAAGAAGAUGCUAAACGUdTdT in Fig. 5f and Supplementary Fig. 8b, d or M-006626-01 in Fig. 5e).

**Immunofluorescence microscopy**. HeLa cells were seeded on coverslips in a 6-well plate. For staining of chromatin-bound Pol II, cells were extracted as described previously[70] with CSK buffer (10 mM Pipes (pH 6.8), 100 mM NaCl, 300 mM sucrose, 3 mM $MgCl_2$, 0.5% (v/v) Triton X-100) for 3 min at room temperature (RT) and washed twice in PBS. Cells were then fixed with 2% (v/v) formaldehyde at RT for 12 min and washed three times before incubation with Pol

II 8WG16 (Abcam, ab817) in PBS with 5% (v/v) FBS for 1 h. Cells were washed twice, incubated for 30 min at RT with donkey anti-mouse IgG (H + L) highly cross-adsorbed secondary antibody Alexa Fluor 488 (ThermoFisher, A-11001) and washed three times in PBS. For BrdU staining, cells were incubated with 10 µM of BrdU for 1 h (Supplementary Fig. 1a) or 100 µM for 30 min (Supplementary Fig. 2c) and the incorporated BrdU was detected using mouse anti-BrdU (clone B44; BD Biosciences) as described in ref. [70]. Coverslips were mounted using Fluoromount-G (SouthernBiotech, 0100-01) containing 4′,6-diamidino-2-phenylindole (DAPI) and slides were visualized with an inverted confocal microscope (LSM 880; ZEISS). Fluorescence intensity was analyzed with ImageJ (version 1.52p) as described previously[70]. Pol II signal per nucleus was represented by using box-and-whisker plot with GraphPad Prism 8.3.1 software with the following settings: boxes: 25–75 percentile range; whiskers: 10–90 percentile range; horizontal bars: median. Micronuclei scoring was performed in HeLa cells plated in 96-well plates (CellCarrier; PerkinElmer). For automated counting, 96-well plates were scanned with a 20X objective using an Operetta High-Content Imaging System (PerkinElmer) with Harmony software (version 4.8). Analysis was carried out by defining

**Fig. 5 R-loops contribute to activation of inflammatory response in RNase H2-depleted cells. a** ChrRNA-seq profiles for *INFNGR1* (top), *PTGS2* (middle) and *TUBG2* (bottom) genes in HeLa cells transfected with siLuc (blue) and siRNASEH2A (red). *TUBG2* is a control gene. Each track represents an average signal from two biologically independent experiments (*n* = 2). **b** qRT-PCR of indicated mRNAs in HeLa cells transfected with siLuc (white bars) or siRNASEH2A (green bars). Values are relative to siLuc cells (means ± SEM); *n* = 8 (*OAS1, PTGS2, ISG20, STING*), *n* = 5 (*IFNGR1, TNF, TUBG2*), *n* = 3 (*SELENBP1*) biologically independent experiments. *p*-values were calculated using two-tailed unpaired *t*-test. **c** HEK293T RNase H1-FLAG-IRES-mCherry cells transfected with siLuc (white bars) or siRNASEH2A (green bars) and induced with doxycycline (DOX) to over-express RNase H1 (shades of blue) as in Fig. 4g. qRT-PCR analysis of indicated mRNAs. Values are relative to siLuc or siLuc +DOX (means ± SEM); *n* = 6 (*IFNGR1, PTGS2, TNF*), *n* = 3 (*TUBG2* and *SELENBP1*) biologically independent experiments. *p*-values were calculated using two-tailed unpaired *t*-test. **d** Alkaline comet assays in HEK293T RNase H1-FLAG-IRES-mCherry cells treated as in **c**. Representative images (left) and quantification of comet tail moment (right). Scale bars: 100 μm. Boxplot settings are: box: 25–75 percentile range; whiskers: 10–90 percentile range; horizontal bars: median; outliers not displayed. >500 nuclei were quantified per condition. Representative data from *n* = 4 biologically independent experiments. *p*-values: **$p$ = 0.0065, ****$p$ < 0.0001 (one-way ANOVA, Tukey's multiple comparison test). **e** Alkaline comet assays in HeLa cells transfected with the indicated siRNAs (siXPG are in shades of orange, siXPF in shades of purple). Representative images (left) and quantification of comet tail moment (right). Scale bars: 100 μm. Boxplot settings are: box: 25–75 percentile range; whiskers: 10–90 percentile range; horizontal bars: median; outliers not displayed. Representative data from *n* = 3 biologically independent experiments. >400 nuclei were quantified per condition. ****$p$ < 0.0001 (one-way ANOVA, Tukey's multiple comparison test). **f** qRT-PCR analysis of indicated mRNAs (siRNASEH2A alone is in green, siXPG are in shades of orange). Some siRNA transfections were performed in parallel to siAQR and siXPF and have the same siLuc and siRNASEH2A controls as in Supplementary Figs. 7e and 8f. Values are relative to siLuc and siXPG cells (means ± SEM); *n* = 3 (*IFNGR1, TUBG2, SELENBP1*), *n* = 5 (*TNF*) and *n* = 7 (*PTGS2*) biologically independent experiments. *p*-values were calculated using two-tailed unpaired *t*-test. Source data are provided as a Source data file.

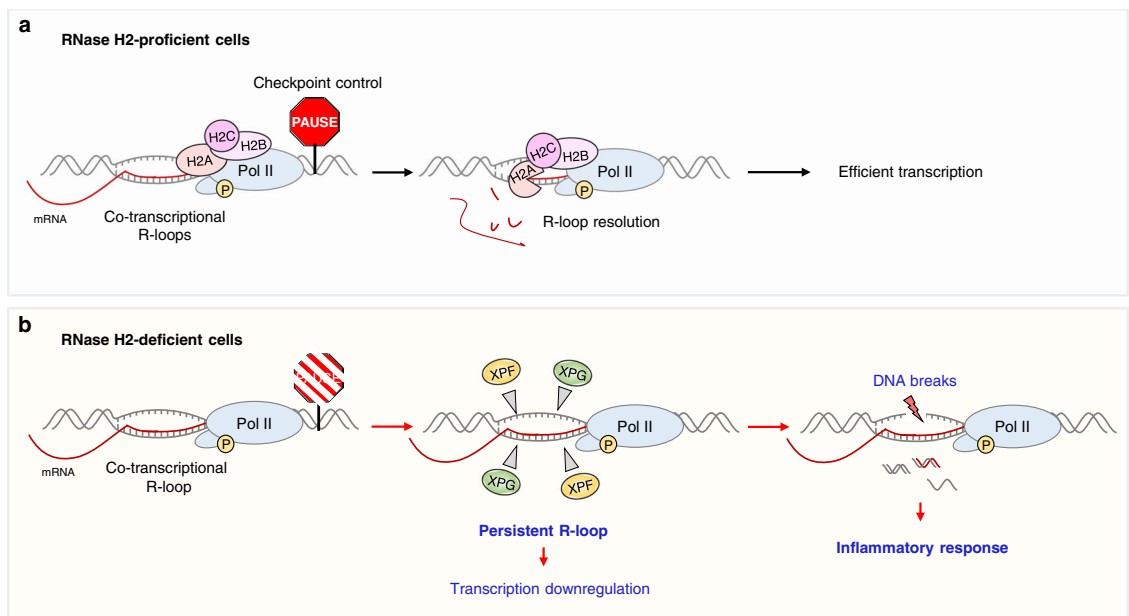

**Fig. 6 Model for the role of RNase H2 complex in R-loop resolution during transcription. a** RNase H2 binds actively transcribed genes and interacts with phosphorylated Pol II. If co-transcriptional R-loops exceed the physiological levels, RNase H2 resolves them, ensuring a checkpoint control to promote efficient transcription. **b** Lack of RNase H2 results in the lack of this checkpoint control and accumulation of R-loops at short genes, impaired transcription, XPF- and XPG-mediated R-loop cleavage and consequent DNA break formation which activates an inflammatory response, characteristic of AGS pathology.

a ring region around the nuclei and identifying the micronuclei in this region by using the "B" method with Columbus software (version 2.8.2). Antibodies used in this study can be found in Supplementary Information file.

**5′-EU labeling.** HeLa cells were seeded on coverslips and incubated with 1 mM 5-ethylnyl uridine (EU) for 1 h to label newly synthesized RNA. For automated EU analysis HeLa and HEK293T cells were seeded in 96-well plates (CellCarrier; PerkinElmer). 96-well plates were coated with poly-L-lysine (Sigma-Aldrich) for HEK293T. Global nascent RNA transcription was detected using Click-iT RNA Alexa Fluor 488 Imaging Kit (ThermoFisher, C10329) according to the manufacturer's instructions. Following Click-iT reaction cells were immunostained by incubation with an anti-nucleolin antibody in 2% (v/v) BSA in PBS for 45 min at RT. After two washes, donkey anti-rabbit IgG (H + L) highly cross-adsorbed secondary antibody Alexa Fluor 594 (Life Technologies, A-21207) or goat anti-rabbit IgG (H + L) highly cross-adsorbed secondary antibody Alexa Fluor 647 (Life Technologies, A-21245) was added for 1 h at RT and washed two times in PBS. Coverslips were mounted using Fluoromount-G (SouthernBiotech, 0100-01) containing DAPI and slides were visualized with an inverted confocal microscope

(LSM 880; ZEISS). Fluorescence intensity was analyzed with ImageJ (version 1.52p) as described previously[70]. In 96-well plates, nuclei were stained with 1 μg/ml Hoechst 33342 for 15 min, washed twice with PBS and stored at 4 °C until analysis. 96-well plates were scanned with a 20X objective using an Operetta High-Content Imaging System (PerkinElmer) with Harmony software (version 4.9). Analysis and quantifications were carried out by defining the nucleoli by using the "B" method with Columbus software (version 2.8.2). Data were plotted by using box-and-whisker plot with GraphPad Prism 9.1.0 software with the following settings: boxes: 25–75 percentile range; whiskers: 10–90 percentile range; horizontal bars: median.

**Cellular fractionation and immunoblotting.** Whole cell extracts (WCE) were prepared by lysing the cells for 20 min on ice in RIPA buffer (25 mM Tris-HCl pH 7.6, 150 mM NaCl, 1% NP-40, 1% Sodium Deoxycholate, 0.1% SDS) supplemented with 1X Complete EDTA-free protease (Sigma-Aldrich) and phosphatase (Halt phosphatase inhibitor cocktail; ThermoFisher) inhibitors. Extracts were then briefly sonicated, centrifuged and supernatant was collected. Chromatin-bound proteins were isolated as described previously[70]. Briefly, cells were lysed twice in extraction buffer (50 mM Hepes (pH 7.5), 150 mM NaCl, 1 mM EDTA) containing

0.1% (v/v) Triton X-100 supplemented with 1X Complete EDTA-free protease and phosphatase inhibitors for 15 min at 4 °C. 'Soluble protein fraction' (supernatant) was collected after centrifugation at $14,000 \times g$ for 3 min. The pellet was resuspended in extraction buffer without Triton X-100 supplemented with 200 µg/ml RNase A (PureLink, Invitrogen) for 20 min at 25 °C under agitation. The "RNase A fraction" was collected following centrifugation at $14,000 \times g$ for 3 min. Insoluble pellet was resuspended in 1% SDS and 10 mM Tris-HCl pH 7.4 supplemented with 1X Complete EDTA-free protease and phosphatase inhibitors and sonicated. Proteins were separated on 4–12% Bis-Tris, 3–8% Tris-Acetate (Invitrogen) gels or 12.5% SDS-PAGE and immunoblotted with indicated antibodies. Immunoblotting was revealed by chemiluminescence using autoradiography or a ChemiDoc MP Imaging System (Bio-Rad). Protein level was quantified using Image Studio Lite or ImageJ for autoradiography and Image Lab for ChemiDoc-acquired images. Antibodies used in this study can be found in Supplementary Information file.

Full scans of blots are provided in Source data files.

**Comet assays.** Neutral and alkaline comet assays were performed according to the manufacturer's protocol (Trevigen) by performing electrophoresis at 4 °C. Slides were visualized with a AxioObserver Z1 fluorescence microscope (ZEISS) with the objective EC Plan-Neofluar 10X/0.3 Ph1. Comet tail moments were analyzed with ImageJ software (version 1.52p) automatically with the plugin OpenComet (http://www.cometbio.org) or manually using a macro provided by Robert Bagnell (https://www.med.unc.edu/microscopy/resources/imagej-plugins-and-macros/comet-assay) as previously described[70]. Data were represented by using box-and-whisker plot with GraphPad Prism 8.3.1 or 9.1.0 software with the following settings: boxes: 25–75 percentile range; whiskers: 10–90 percentile range; horizontal bars: median. In Supplementary Fig. 8c, data were represented by using scattered dot blot with the horizontal bar indicating the mean.

**Co-immunoprecipitation (Co-IP).** BSA-blocked protein A dynabeads (Invitrogen) were incubated with RNase H2A, RNase H2C or matched isotype IgG antibodies in IP buffer (25 mM Tris-HCl pH 8.0, 150 mM NaCl, 0.5% NP-40%, 10% glycerol, 2.5 mM MgCl$_2$) supplemented with 1X Complete EDTA-free protease inhibitors (Sigma-Aldrich) at 4 °C, for 2 h. Pelleted cells were incubated with lysis buffer (50 mM Tris-HCl pH 8.0, 150 mM NaCl, 1% NP-40, 10% glycerol, 2.5 mM MgCl$_2$) supplemented with 0.5 mM PMSF, 1X Complete EDTA-free protease (Sigma-Aldrich) and 1X phosphatase (PhosSTOP, Roche) inhibitors in the presence of 40 U benzonase (E1014, Sigma-Aldrich) at 4 °C for 30 min. Insoluble material was removed by centrifugation and supernatant was diluted 1:2 in dilution buffer (150 mM NaCl, 10% glycerol, 2.5 mM MgCl$_2$) supplemented with 0.5 mM PMSF, 1X Complete EDTA-free protease (Sigma-Aldrich) and 1X phosphatase (PhosSTOP, Roche) inhibitors. 1 mg of proteins were subjected to IP with antibody-conjugated beads at 4 °C for 2 h. Immuno-complexes were washed three times with IP buffer and three times in IP buffer without NP-40, both supplemented with 1X Complete EDTA-free protease inhibitors (Sigma-Aldrich). Immuno-precipitated complexes were eluted in 1X LDS (Invitrogen), 100 mM DTT for 10 min at 70 °C and proteins were resolved on 4–12% Bis-Tris or 3–8% Tris-Acetate (Invitrogen) by SDS-PAGE and immunoblotted with indicated antibodies. Immunoblotting was revealed by chemiluminescence using autoradiography or a ChemiDoc MP Imaging System (Bio-Rad). Antibodies used in this study are listed in Supplementary Information file.

**Chromatin immunoprecipitation (ChIP).** Chromatin immunoprecipitation was performed as described previously[53]. Briefly, cells were crosslinked with 1% formaldehyde at 37 °C for 15 min and the reaction was quenched with 0.125 M glycine at 37 °C for 5 min. Nuclei were isolated by centrifuging cells after lysis (5 mM PIPES pH 8.0, 85 mM KCl, 0.5% NP-40 and 1X Complete EDTA-free protease inhibitors, Sigma-Aldrich). Nuclei were lysed in nuclear lysis buffer (50 mM Tris-HCl pH 8.0, 0.5 mM EDTA, 1% SDS, 0.5 mM PMSF, 1X Complete EDTA-free protease inhibitors, Sigma-Aldrich) and sonicated (Diagenode Bioruptor). Insoluble material was removed by centrifugation and samples were diluted in ChIP IP buffer (16.7 mM Tris-HCl pH 8.0, 1.2 mM EDTA pH 8.0, 167 mM NaCl, 0.01% SDS, 1.1% Triton-X-100, 0.5 mM PMSF, 1X Complete EDTA-free protease inhibitors, Sigma-Aldrich). Samples were precleared by incubation with protein A dynabeads (Invitrogen) blocked with BSA (B8894, Sigma-Aldrich) or with protein A agarose beads (16–157; Millipore). Precleared chromatin was incubated overnight at 4 °C with Pol II, RNase H2A, RNase H2C or no antibody. Samples were incubated with protein A agarose beads (16–157: Millipore) or with BSA-blocked protein A dynabeads to collect the immuno-complexes. Beads were washed once with buffer A (20 mM Tris-HCl pH 8.0, 2 mM EDTA, 0.1% SDS, 1% Triton X-100 and 0.15 M NaCl), once with buffer B (20 mM Tris-HCl pH 8.0, 2 mM EDTA, 0.1% SDS, 1% Triton X-100 and 0.5 M NaCl), once with buffer C (10 mM Tris-HCl pH 8.0, 1 mM EDTA, 1% NP-40, 1% Sodium Deoxycholate and 0.25 M LiCl) and twice with buffer D (10 mM Tris-HCl pH 8.0 and 1 mM EDTA). Immuno-precipitated complexes were eluted with 1% SDS and 0.1 M NaHCO$_3$. Samples were de-crosslinked by adding RNase A and 0.3 M NaCl (final concentration) at 65 °C, at least 4 h, and treated with proteinase K (Sigma-Aldrich) at 45 °C, 2 h. DNA was purified with QIAquick PCR purification kit (QIAGEN) and analyzed by qPCR with Rotor-Gene® Q (QIAGEN) using QuantiTect SYBR green. CFX96™ Real-

time System (Bio-Rad) with IQ™ SYBR® Green Supermix (Bio-Rad) and corresponding analysis software (Bio-Rad CFX Manager 3.1) was used for Supplementary Fig. 6c, d. The amount of immunoprecipitated material at a particular gene region was calculated as the percentage of input after subtracting the background signal (no antibody control). The primers used are shown in Supplementary Table S1. For ChIP-Sequencing analysis, multiple ChIP IPs were pooled. DNA was purified with MinElute® PCR purification Kit (QIAGEN) and subjected to library preparation and sequencing on a HiSeq4000 with 75 bp paired ends reads at Oxford Genomics Centre (WTCHG, University of Oxford). The sequences of the primers used in this study can be found in Supplementary Table 1.

**Chromatin-bound RNA (ChrRNA)-seq.** ChrRNA method was performed as described previously[40]. Briefly, $7 \times 10^6$ cells for each condition were collected in ice cold PBS. Cells were pelleted at $420 \times g$ for 5' at 4 °C and incubated in 4 ml of HLB/NP40 buffer (10 mM Tris-HCl pH 7.5, 10 mM NaCl, 0.5% NP40 and 2.5 mM MgCl$_2$) for 5 min. 1 mL of ice-cold HLB/NP40 buffer with addition of 10% sucrose was under-layered and nuclei were isolated by centrifugation for 5 min at $420 \times g$ at 4 °C. Isolated nuclei were resuspended in 125 µl of NUN1 solution (20 mM Tris-HCl pH 8.0, 75 mM NaCl, 0.5 mM EDTA, 50% Glycerol) supplemented with 1X Complete EDTA-free protease inhibitors (Sigma-Aldrich). Resuspended nuclei were incubated on ice in 1.2 mL of NUN2 buffer (20 mM HEPES-KOH pH 7.6, 7.5 mM MgCl$_2$, 0.2 mM EDTA, 300 mM NaCl, 1 M Urea, 1% NP40) supplemented with 1X Complete EDTA-free protease inhibitors (Sigma-Aldrich) for 15 min. RNA-bound chromatin was pelleted for 10 min at 4 °C and the DNA was digested by incubating the chromatin pellet at 37 °C for 15 min shaking in 200 µL HSB (10 mM Tris-HCl pH 7.5, 500 mM NaCl and 10 mM MgCl$_2$) with 0.25 U/µL TURBO DNase (ThermoFisher, AM2238). Proteins were digested by adding 200 µL of proteinase K solution (0.4% SDS and 0.8 mg/mL Proteinase K, Sigma-Aldrich) at 37 °C for 10 min. Chromatin-bound RNA was extracted in TRIzol Reagent (ThermoFisher, 15596018) according to the manufacturer guidelines. Chromatin-bound RNA was dissolved in water and RNA integrity was checked on the Agilent 4200 TapeStation system (Agilent Technologies). 5 µg of chromatin-bound RNA was depleted of ribosomal RNA with RiboMinus Eukaryote System v2 (Thermo Fisher, A15015) according to manufacturer's protocol. Library preparation with NEBNext Ultra II Directional RNA Library Prep kit for Illumina (E7760) and sequencing on a NovaSeq 6000 with 150 bp paired ends reads was performed at Oxford Genomics Centre (WTCHG, University of Oxford).

**DNA/RNA immunoprecipitation (DRIP).** DNA/RNA immunoprecipitation was performed as described previously[53]. Briefly, non-crosslinked nuclei were lysed in nuclear lysis buffer (50 mM Tris-HCl pH 8.0, 5 mM EDTA, 1% SDS) and digested with proteinase K (Sigma-Aldrich) at 55 °C for 3 h. Genomic nucleic acids were precipitated with isopropanol, washed in 75% ethanol and sonicated with Bioruptor (Diagenode) in IP dilution buffer (16.7 mM Tris-HCl pH 8.0, 1.2 mM EDTA, 167 mM NaCl, 0.01% SDS, 1.1% Triton X-100). Samples were precleared in the presence of protease inhibitors (0.5 mM PMSF, 0.8 mg/ml pepstatin A, 1 mg/ml leupeptin) with protein A dynabeads (Invitrogen) blocked with BSA (B8894, Sigma-Aldrich). 10 µg of precleared genomic DNA were incubated overnight at 4 °C with S9.6 antibody or no antibody. RNase H digestion was performed by incubation with 1.7 U RNase H (M0297, NEB) per µg of genomic DNA for 2.5 h at 37 °C before IP. Incubation with beads, washes and elution steps were performed in the same way as described for ChIP. Samples were digested with proteinase K (Sigma-Aldrich) at 45 °C for 2 h, DNA was purified with QIAquick PCR purification kit (QIAGEN) and analyzed by qPCR with Rotor-Gene® Q and Quanti-Tect SYBR green (QIAGEN). At a certain gene region, the amount of immuno-precipitated material was calculated as the percentage of input after subtracting the background signal (no antibody control). The primers used are shown in Supplementary Table S1. For DRIP-seq analysis, multiple S9.6 IPs were pooled. DNA was purified with MinElute® PCR purification Kit (QIAGEN) and subjected to library preparation and sequencing on HiSeq 4000 with 75 bp paired ends reads at Oxford Genomics Centre (WTCHG, University of Oxford). The sequences of the primers used in this study can be found in Supplementary Table 1.

**RNA analysis.** Total RNA was isolated using TRIzol Reagent (ThermoFisher, 15596018) according to the manufacturer guidelines followed by DNase I treatment (Sigma-Aldrich, 04716728001) at 37 °C for 2 h. cDNA was obtained by reverse transcription of 1–2 µg of total RNA using SuperScript Reverse Transcriptase III (Invitrogen) with random hexamers (ThermoFisher) and analyzed by qPCR with Rotor-Gene® Q (QIAGEN) and QuantiTect SYBR green (QIAGEN). Values are normalized to *GAPDH* mRNA, used as control. The sequences of the primers used in this study can be found in Supplementary Table 1.

**Statistics and reproducibility.** Information on biological replicates ($n$) is indicated in the figure legends. Unless otherwise stated, experimental differences for IF, DRIP, ChIP and RT-qPCR were tested for significance with two-tailed unpaired $t$-test. One-way ANOVA with correction for multiple comparisons (Tukey test) was used to calculate statistical significance in comet experiments. For sequencing data, significance was calculated by using Wilcoxon rank sum test, Kruskal-Wallis test and paired Wilcoxon signed rank test. Statistical tests were performed using GraphPad

Prism 8.3.1 or 9.1.0 software. *p*-values are indicated in the figure legends. $p \leq 0.05$ is considered significant. *$p \leq 0.05$, **$p \leq 0.01$, ***$p \leq 0.001$, ****$p \leq 0.0001$.

**Reporting summary**. Further information on research design is available in the Nature Research Reporting Summary linked to this article.

## Data availability

ChIP-seq, DRIP-seq, chrRNA-seq, and the bed files for called peaks generated in this study have been deposited in NCBI's Gene Expression Omnibus (GEO) and are accessible through GEO Series accession number GSE146970. Source data are provided with this paper.

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

## Acknowledgements

We thank the Oxford Genomics Centre at the Wellcome Centre for Human Genetics (funded by Wellcome Trust grant reference 203141/Z/16/Z) for the generation and initial processing of the sequencing data. We gratefully thank the Micron Advanced Bioimaging Unit (supported by Wellcome Strategic Awards 091911/B/10/Z and 107457/Z/15/Z) and the CRCT Cell Imaging Platform (Cancer Research Center of Toulouse) for the support and assistance with microscopy. We thank N. Proudfoot for critically reading the manuscript. We thank T. Nojima, L. Palazzo, K. Kamieniarz-Gdula and G. Dujardin for technical advice. N.G lab is supported by a Royal Society University Research Fellowship (UF150656), EPA Research Fund (Sir William Dunn School of Pathology, University of Oxford), an MRC New Investigator Research Grant (MR/J007870/1), a John Fell award (BVD07340, Ref 133/090) to N.G. and Medical Sciences Internal Fund (University of Oxford, Ref 0005681) to A.C. M.T. and S.M. are supported by Wellcome Trust Investigator Awards WT106134AIA and WT210641/Z/18/Z to S.M. OS lab is supported by the Fondation pour la Recherche Médicale (FRM) [Equipe labellisée FRM (DEQ20170839117).

## Author contributions

A.C. designed and performed experiments, analyzed data unless otherwise noted. M.T. and S.M. performed bioinformatics analysis. F.C. performed RT-qPCR, WB and analyzed the data in Fig. 5; WB and DRIP-qPCR in Fig. 4g, h. C.A. performed ChIP-qPCR, DRIP-qPCR and WB. L.O.A. generated initial ChIP-qPCR and co-IP data. N.B. generated HEK293T-Tet-inducible RNase H1-FLAG-IRES-mCherry. R.H. performed co-IP experiments. O.S. contributed to data analysis. N.G. designed and supervised the project, analyzed the data. A.C. and N.G. wrote the manuscript with contribution from other authors.

## Competing interests

The authors declare no competing interests.
