## [Peer Review File · Nature Communications]

REVIEWER COMMENTS

Reviewer #1 (Remarks to the Author):

RNase H2, the subject of these studies, has two different functions. The most studied, at least in mammals, is the ribonucleotide excision repair (RER) activity, which initiates the removal of ribonucleotides incorporated in DNA during replication. This manuscript addresses the least known function of RNase H2 in processing the RNA strand of R-loops, which are structures formed after transcription when the RNA hybridizes back with the template strand. This work describes the association of RNase H2 to actively transcribed genes, analyzing RNase H2 binding to chromatin and its genomic distribution. An association to chromatin could be due to its RER activity, but they found the interaction to be independent of DNA replication by inhibiting DNA Pol and by using quiescent fibroblasts. They found that RNase H2 and RNA Pol II co-IP in the absence of nucleic acids, suggesting that they are part of the same complex. When RNase H2 is depleted, nascent transcripts are reduced for some gene categories but not (or very minimally) for intron-containing genes, which make the largest portion of RNA Pol II transcripts. Consistent with RNase H2 interacting and processing the R-loops of only a small subset of active genes, DRIP-seq showed more decreased/loss (35,604) than increased/gained (16,647) DRIP peaks in RNase H2 depleted cells, although selecting for genes that are strongly enriched for R-loops, there is an increase in R-loop signal upon RNase H2A depletion in most gene categories except in intron-containing genes. Finally, they try to establish a connection between R-loop accumulation in the absence of RNase H2 and DNA damage and the immune response found in AGS patients.

The main findings of the manuscript are:

- RNase H2 interacts with RNA Pol II and acts co-transcriptionally removing R-loops presumably as they are formed to allow proper gene expression.
- There is a specific subset of genes that RNase H2 binds, leading to R-loop processing. This group of RNase H2-interacting genes are mostly short genes including histone, snRNA and intronless genes.
- They propose a mechanism for R-loop induced DNA damage and immune-response in RNase H2 defective and AGS mutants that requires structure-specific endonucleases cleaving the ssDNA portion of the R-loops.

All these results are novel and would be of interest to the fields of RNase H studies, DNA damage and autoinflammatory disorders. Also, they would advance our understanding of how R-loop accumulation affects gene regulation. The data are well presented, and the statistical analyses are appropriate.

However, before been accepted for publication a few issues should be addressed:

1. RNASEH2C mRNA is only depleted to about 60% by siRNASEH2C (Extended data figure 1C) and subsequently the protein level of RNase H2C is only reduced to about 50% (Figure 1b) of wt levels forming RNase H2 complexes that are about 50% the amount in wt cells. These small decreases would presumably affect all subsequent data obtained using siRNASEH2C. I would suggest using siRNASH2B instead, which appears to be more effective.
2. I would like to see a quantification of the western bands in Extended data Figure 1e to see how it compares to Figure 1a-b. Comparing the Extended Data Figure 1e with Figure 1a, the fraction of chromatin bound RNase H2 subunits appears to be very similar after siLuc as after depletion of the different components of the RNase H2 complex. It seems that the depletion affects mostly the free form

of RNase H2 but shows little effect on the chromatin-bound form, or that the fraction of bound form is very small.

3. Because RNase H2 would be expected to associate with chromatin as part of its RER function, it is important to show convincingly that the association of RNase H2 and RNA Pol II is strong in quiescent fibroblasts, which the data presented in Extended Data Figure 3b does not appear to support well. The inclusion of clear western blot data is needed to help substantiate this association.

4. In figure 4b it is shown that RNase H2 peak coincides with the DRIP peak in cells with wt RNase H2, suggesting that RNase H2 binds but doesn't cleave R-loops enriched regions. Could an explanation for this be proposed? Perhaps RNase H2 binds all or most R-loops but only processes a small portion of them.

5. That the depletion of RNase H2 induces immune-related mRNAs is something that has been described in several systems, and it was previously shown to be associated with its RER activity. To demonstrate an involvement of the R-loop processing activity of RNase H2, the authors used overexpression of RNase H1. They showed that increased RNase H1 activity decreased the expression of immune-related genes. However, overexpression of RNase H1 may have additional effects, such as decreasing mtDNA replication. In addition, it has not been established whether RNase H1 can functionally replace RNase H2 in processing R-loops in mammalian cells. A much better experiment would be to delete the RNASEH2A gene and express the RNaseH2A-RED mutant, which has been successfully used for separating the two activities of RNase H2 and is specific for the removal of R-loops.

6. Does the RNase H1 overexpression system express only the nuclear form of the enzyme, or does it also express the mitochondrial form? This construct should be clearly described.

Reviewer #2 (Remarks to the Author):

This manuscript describes the genome-wide profile of RNaseH2 binding to the human genome. The main findings are that RNaseH2 associated with transcribed regions of the genome in a transcription-dependent manner and through physical association with RNA polymerase II. Loss of RNaseH2 function leads to defects in transcription that appear selective to specific types of transcripts, namely short intronless transcripts. As expected, loss of RNaseH2 leads to increases in R-loops at some loci. These R-loops can lead to DNA damage caused by the XPF/XPG nucleases and activate transcription of inflammatory response genes, which is relevant to the RNaseH2-deficiency disorder Aicardi-Goutieres syndrome. Overall, the manuscript is clear and straightforward and presents a novel finding that RNaseH2 regulates transcription through association with RNA polymerase II and transcribed genes.

I have a number of technical questions, and suggestions for additional analyses that could improve the work. In short, while Figures 1-3 are novel, Figures 4 and 5 primarily serve to validate the literature but do not add much new insight. In the end this leads to a model which lacks some key insights; specifically how and when does RNaseH2 interact with RNA polymerases? how does rapid degradation of a co-transcriptional R-loop promote efficient transcription, and what does this have to do with short genes? How do these R-loops trigger an inflammatory response?

A deeper exploration of some of these questions, even using the authors existing data and some assays

they already have working would dramatically strengthen the manuscript.

Major issues:

1. The EU staining data seems to also show that nucleolar intensity is decreased. The authors should test whether RNaseH2 also affects RNA Polymerase I and III transcription, and whether RNA polymerase I and III can be observed in pulldowns of RNaseH2 (as in Fig. 3A). This is a missed opportunity to learn more about how RNaseH2 is actually recruited. For example, if all three polymerases could recruit RNaseH2 then it would suggest that a shared subunit may be responsible for the interaction.

2. The correlation between R-loop levels and RNaseH2 occupancy is quite strange. Naively one would expect higher RNaseH2 at a locus to reduce the lifetime and amount of R-loops present. The authors suggest other factors may compensate. I wonder if catalytic mutants of RNaseH2A would behave the same way in terms of their effects on transcription. The authors could attempt to rescue their siRNA cells with WT or mutant RNaseH2A to determine if the R-loop resolution function is required for the observed effects on global transcription.

3. The induction of immune related genes by qPCR Figure 5f seem to normalize both the siLuc and siXPG to 1 for the mRNA levels. I think the siXPG should not be normalized and should be compared directly to siLuc. Does XPG or XPF knockdown alone increase inflammatory gene expression? It may be that the relative increase in siRNaseH2A is lower, but if the background in siXPG is much higher it could change the interpretation. The way the data is presented does not allow for this kind of comparison.

4. In Figure 5e, Might the authors have expected that XPF depletion on its own would have caused DNA damage. The box plot should perhaps show the individual data points, it seems like the siXPF sample has a long tail of cells with damage. Is the distribution normal? Can the authors do a statistical test of siXPF versus control to see if there is an increase in comet tail moment? Currently the siXPF bar (purple) is not compared to the control siRNA bar (white) as far as I can see. This should be done and discussed. Which post-hoc statistical test was used should be indicated in the legend also. In addition, if the damage is really due to R-loops RNaseH1 expression could be included as an additional control, especially since XPF/XPG knockdown do not completely suppress the damage.

5. Is it possible that the immune genes are directly affected by RNaseH2A knockdown due to the transcriptional role? To address any effects the authors could analyze the DRIP, RNAPolIII and RNaseH2A occupancy at all of the reporter genes (TNF, STING etc). This is important to rule out direct effects on transcription as opposed to DNA damage-cytoplasmic DNA induced activation. Similarly, the authors rely on mRNA expression but looking at protein level induction and possibly at micronuclei as a cause of STING activation would fill out the story. Additional data to help us understand how these R-loops trigger an inflammatory response would help better justify the focus on AGS in the title.

6. The bias toward short genes is not really explained because RNaseH2 and DRIP signal still occupy a lot of normal genes. If it is only when RNaseH2 is depleted that the effect on short genes is seen, then why does it RNaseH2 occupy all of those other genes. This has got to be explained better. See my comment above (#2) on testing whether the R-loop resolution activity is even required for the effects on transcription. The identity of genes where R-loops are lost in RNaseH2-knockdown would be another

interesting place to reanalyze and get additional insights.

- Later on Page 7 the authors hint at a very exciting model for why there are redundant pathways at long genes. Reading between the lines are you saying if a short gene forms an R-loop and gets 'stuck' the cell can afford to just make it over again, so degradation by RNaseH2 is preferred. For very long genes this is a bad strategy because degradation of that long transcript wastes a lot of energy and time to remake the long transcript, so other pathways predominate? I'd like the authors to confirm and expand on this model if they agree. Could analysis of DRIP peak changes in other siRNA conditions confirm that RNaseH2-effects short genes while other pathways affect long genes?

- This paper (<https://pubmed.ncbi.nlm.nih.gov/32747416/>) generates DRIP data in a few other cell lines with other siRNAs, maybe the gene length effects could be analyzed in these data sets and compared with the gene length of DRIP-gain genes RNaseH2A?

7. I do not really think the model is explained well. How does R-loop degradation lead to efficient transcription? It would seem that it would degrade the transcript, how does this help efficiency? The authors may need to consider literature on stalled RNA polymerase and associated R-loops. Could the stabilized R-loops reflect a backlog of stalled polymerase that needs to be cleared for productive transcription?

Minor issues:

- Page 4. The concluding sentence of the section "Taken together, our results imply that RNase H2 plays a role in gene transcription" is overstated. At this point in the story, the authors just finished show that RNaseH2 is recruited to the genome in a transcription dependent manner, it does not imply that RNaseH2 has a role in transcription.

- In figure 1A, siRNaseH2A shows complete KD. In figure 5C siRNaseH2A is not as good. I recognize that the cell lines are different but the differences are quite striking. Could this low penetrance of knockdown in Figure 5 have affected the phenotype?

- Can you justify use of the fibroblast model versus a neuronal model system. A more relevant model system would strengthen the arguments. Indeed, the title of the paper really suggests a focus on AGS pathology, which is not well developed here.

- Information about what constitutes a gained DRIP peak or a lost DRIP peak should be included in the main text on Page 5. A reader will see that the total number of peaks does not add up with the gained/lost peaks. I assume this is due to a threshold of changes in peak reads but it would help to clarify exactly what the authors consider a gained/lost peak.

- The authors state in places that RNaseH2 forms a complex with RNA polymerase II. The only evidence shown is that they interact by co-IP and western. I think the word 'complex' may be overstating the results. We do not learn a lot in this study about when and how RNaseH2 interactions with RNA polymerase II occur.

- The rationale for using an alkaline comet assay in Figure 5 is unclear. Are the authors expecting double strand breaks? If not, then how could ssDNA breaks trigger an inflammatory response? A neutral comet assay might help them better connect the damage to STING activation.
- The legend for Figure 2C says ActD is compared to a DMSO vehicle but the figure says EtOH. This should be clarified.
- The legend for Figure 2 also refers to panel g, as (d).
- The legend for Figure 1 labels intronless JUNB as k, while it should be l.

Reviewer #3 (Remarks to the Author):

This is a nice paper showing a kind of unexpected role of RNASEH2 during transcription by RNA polIII. Using DRIP-seq and ChIP-seq to analyze DNA-RNA hybrids and the presence RNA polIII and RNASEH2A, the authors show an enrichment of RNASEH2A at the 5' end region of genes. Removal of RNASEH2 leads to an increase of hybrids mainly at the regions where RNASEH2A is enriched in normal cells that, as expected from previous studies, is associated with DNA damage. At least part of this damage is produced by structure-specific endonucleases, which is one mechanism of break formation. Consistent with the current literature, damage leads to an enhancement in the RNA levels of genes of the immune response. In general, the manuscript provides new and interesting results that make it a candidate for Nat Comm. I have few comments added below to improve the manuscript, but also some important requests. It seems that the manuscript provides ChIP-seq and DRIP-seq data performed only once. At least the DRIP-seq and ChIPseq of RNH2A should be performed minimally twice to show that results are repetitive, to make this a Nat Comm article. In addition the model proposed has to be better justified. It is pretty counterintuitive and the data are open to additional interpretations.

Specific comments

It would be good to justify why they chose for DRIP-qPCR of related genes RNU1 and RNU2

Rather than showing the metaplot of intron-plus versus intron-less genes, which really reflects a comparison of short versus long genes, it would be more informative to see the data of ChIP-seq of exons versus introns.

Authors cannot claim that RNASEH2 interacts with RNAPolIII as in page 5: "our results reveal that RNASEH2 is a part of the PolIII complex which acts to promote transcription". For such an affirmation authors would need to show the purification by affinity chromatography and to identify the proteins by MALDI-TOFF, and not just by Westerns of co-IPs. The use of benzonase is not sufficient to conclude that RNASEH2 associates with PolIII in a stable manner regardless of DNA with the results provided. The data indicates that RNASEH2 associates with RNA polIII, but not that it forms a complex with it.

The conclusion that RNASEH2 resolves co-transcriptional R loops needs to be further substantiated. Authors should produce a cell line expressing a catalytically-dead RNASEH2 to show that the nuclease activity is required for R loop resolution. This can be done by RT-qPCR in several of the genes detected to accumulate R loops in the DRIPseq. Alternatively, authors might consider the possibility of reducing R loops by overexpressing the three subunits of RNH2 together

Authors should test whether RNASEH2 is recruited at regions accumulating R loops. Check at some of the RNH2 peaks whether this is reduced by RNASEH1 overexpression. This can be done in several regions by ChIP-qPCR. Wouldn't this be expected if RNASEH2 resolves those hybrids?

I find the last point poorly connected with the rest of the ms. Certainly, the inflammatory response is a highly important phenomena that merits study, but authors should rationalize better how this relates with the rest of the paper. If the authors want to correlate the R loops with DNA damage as a way to strengthen conclusions is OK, but the rest seems to confirm that DNA damage will activate the inflammatory response. This needs to be discussed better. A main question is whether the action of RNASEH2 could be linked to a role in mitochondria or cytoplasm or other events. Can this be excluded?

Fig. 4c with the plot of the DRIP-seq of snRNA data shows a peak downstream the TES. Authors need to discuss this result. Is this supposed to be a non-transcribed DNA region? What is the explanation for a hybrid signal higher than that observed inside the genes?

Fig. 4f. Experiments should show that RNH1 removes the DRIP-qPCR signals in the genes tested.

Fig. 5. Authors need to show the absolute values of mRNA levels, so that the reader can compare different genes. It is easy to double the RNA levels of a gene that is low expressed versus a gene that is highly expressed. This is important considering that results shown correspond to RT-qPCR values and not complete mRNAs. It is critical that the levels of expression of the genes used as controls are similar to those of the immunity response.

Authors should provide a main figure with a large region covering several genes, not just the length of one gene, so that we can compare the ChIP-seq, DRIP-seq, RNASEH2 pChIP-seq at once.

Fig. 5 Considering that results shown correspond to RT-qPCR values, it would be relevant to show a scheme showing which regions has been RT-qPCR to make conclusions.

Fig. 5e. Authors need to show that RNH1 suppresses the comet result, otherwise the result can be explained by the DNA-inserted ribonucleotide removal activity of RNASEH2

I would reconduct the discussion on intron-less and intron-containing genes. The data show that short genes have DRIP-seq signal covering a large part of the gene, whereas intron-containing genes the signal is concentrated on the 5' end. It seems that R loops are accumulated at the 5' end of genes therefore. In long genes, R loops are not seen at the middle and 3' end of genes. With the data observed, it is difficult to think that RNASEH2 travels with the transcription machinery resolving R loops, since almost no effect

or presence of RNASEH2 is observed in the second half of genes longer than 2-kb. It seems its activity is limited only to the 5' end of genes. Intron-less genes and intron-containing genes <2kb shows similar profile indeed. This may be important to discuss in the model, since it may well be possible that RNASEH2 has a role aborting the elongation of suboptimal transcripts that form R loops; otherwise it would be counterproductive for the cell to degrade any long nascent RNA forming short hybrids. Otherwise, how can authors propose that an active RNH2 that would be removing nascent mRNAs forming short stretches of hybrids leads to efficient transcription. Unfortunately authors do not know directionality of the RNA forming the hybrids, since I am not sure whether RNASEH2 could remove hybrids formed with antisense RNA.

Authors should be homogeneous with the plots. Why for instance Ext Fig. 4c, e shows 0.5 kb and +1,5kb region upstream and downstream of genes, whereas g, I show -2 and +2,5 kb. Use in this and all other figures the same length of regions surrounding the genes, so that we can compare all data visually. The same criticism for fig. 1 and 3 metaplots. Indeed, it is not clear to me why the length upstream and downstream is not the same, making thus plots symmetrical.

Detailed Responses to the Reviewers' Comments

We thank the reviewers for their comments and are pleased that they all appreciate the novelty of our findings, in particular the role of RNase H2 function in transcription. In response to the reviewer's comments, we have introduced a number of requested changes, which allowed us to improve the quality and the clarity of our ms. In particular, we have expanded and strengthened the data in our manuscript as follows:

1. We have addressed all technical queries raised by the reviewers with additional controls, experiments and analyses as requested (e.g. **new Fig.3c; 4h; 5d; Supplem Fig. 1e; 3 b-c, d-e; 4 b-c; 6b-d; 7 a, c; 8 a, c, d and g and Response Fig. R1-7**).
2. We have carried out experiments to uncover the relationship between RNase H2 and RNA Polymerases. In particular, using high content microscopy, which allows unbiased automatic detection of thousands of cells, we show that RNase H2A-deficient HeLa and HEK293T cells have transcriptional defects in both nucleoplasm and nucleolus (**new Fig. 3c, new Supplem Fig.3d-e**). In line with this, RNase H2A interacts with RNA Pol I (**new Supplem Fig. 3c**). Finally, RNase H2A forms a complex with transcriptionally engaged RNA Pol II, phosphorylated at Ser2 and Ser5 of the CTD, in non-replicating fibroblasts, supporting the role of RNase H2 in transcription, independently of its function in replication (**revised Supplem Fig. 3b**).
3. We have provided additional mechanistic insight into connection between R-loops and inflammatory response. We now show that RNase H2 depletion triggers both single- and double strand DNA breaks, as measured by alkaline and neutral comet assays, respectively (**new Supplem Fig. 8a**). We demonstrate that accumulated R-loops contribute to DNA break production since RNase H1 overexpression decreases both R-loops and DNA breaks in RNase H2-deficient cells (**new Fig. 4h, Supplem Fig. 6b and Fig. 5d**). Altogether, these new results support a model where R-loops, accumulated due to RNase H2 deficiency especially on short genes, are processed by XPG/XPF endonucleases, which produce DNA breaks, contributing to increased inflammatory response characteristic of AGS pathology.
4. As requested by the Reviewer 2, we have also analysed the function of other R-loops regulators in relation to the gene length (**Response Fig. R4 and ms Discussion**). Our analysis has revealed that activities of some R-loop regulators are influenced by gene length (e.g. TOP1, UAP56/DDX39B and RNase H2) while other factors work independently of gene length (e.g. DDX5, PRMT5 and XRN2). Interestingly, the role of RNase H2 in regulation of co-transcriptional R-loops on short human gene is unique compared to other factors, including TOP1, UAP56/DDX39B, which mostly affect R-loops on long genes.
5. We have provided a better discussion of the working model (**revised Fig.6 and ms Discussion**). We highlight the newly discovered role of RNase H2 in transcription through its interaction with RNA Pol II complex. RNase H2 binds actively transcribed genes and maintains R-loop homeostasis on short genes, by resolving co-transcriptional R-loops which exceed physiological level. In RNase H2-deficient cells, i.e. AGS, these pathological R-loops interfere with transcription and are processed by structure-specific endonucleases XPG/XPF which produce DNA breaks, contributing to inflammatory response.

We hope that you will find our revised ms substantially fortified, providing new insight into how R-loop homeostasis is regulated in human cells and how lack of this regulation impacts gene expression and chronic inflammation. We are also convinced that this study will open future studies in the R-loop, transcription and AGS fields. Please see detailed response to specific comments below.

Reviewer #1

RNase H2, the subject of these studies, has two different functions. The most studied, at least in mammals, is the ribonucleotide excision repair (RER) activity, which initiates the removal of ribonucleotides incorporated in DNA during replication. This manuscript addresses the least known function of RNase H2 in processing the RNA strand of R-loops, which are structures formed after transcription when the RNA hybridizes back with the template strand. This work describes the association of RNase H2 to actively transcribed genes, analyzing RNase H2 binding to chromatin and its genomic distribution. An association to chromatin could be due to its RER activity, but they found the interaction to be independent of DNA replication by inhibiting DNA Pol and by using quiescent fibroblasts. They found that RNase H2 and RNA Pol II co-IP in the absence of nucleic acids, suggesting that they are part of the same complex. When RNase H2 is depleted, nascent transcripts are reduced for some gene categories but not (or very minimally) for intron-containing genes, which make the largest portion of RNA Pol II transcripts. Consistent with RNase H2 interacting and processing the R-loops of only a small subset of active genes, DRIP-seq showed more decreased/loss (35,604) than increased/gained (16,647) DRIP peaks in RNase H2 depleted cells, although selecting for genes that are strongly enriched for R-loops, there is an increase in R-loop signal upon RNase H2A depletion in most gene categories except in intron-containing genes. Finally, they try to establish a connection between R-loop accumulation in the absence of RNase H2 and DNA damage and the immune response found in AGS patients.

The main findings of the manuscript are:

- RNase H2 interacts with RNA Pol II and acts co-transcriptionally removing R-loops presumably as they are formed to allow proper gene expression.
- There is a specific subset of genes that RNase H2 binds, leading to R-loop processing. This group of RNase H2-interacting genes are mostly short genes including histone, snRNA and intronless genes.
- They propose a mechanism for R-loop induced DNA damage and immune-response in RNase H2 defective and AGS mutants that requires structure-specific endonucleases cleaving the ssDNA portion of the R-loops.

All these results are novel and would be of interest to the fields of RNase H studies, DNA damage and autoinflammatory disorders. Also, they would advance our understanding of how R-loop accumulation affects gene regulation. The data are well presented, and the statistical analyses are appropriate. However, before been accepted for publication a few issues should be addressed:

Response: *We would like to thank this reviewer for positive comments and highlighting the novelty of our manuscript and its suitability for publication in Nature Comm journal.*

1. RNASEH2C mRNA is only depleted to about 60% by siRNASEH2C (Extended data figure 1C) and subsequently the protein level of RNase H2C is only reduced to about 50% (Figure 1b) of wt levels forming RNase H2 complexes that are about 50% the amount in wt cells. These small decreases would presumably affect all subsequent data obtained using siRNASEH2C. I would suggest using siRNASH2B instead, which appears to be more effective.

Throughout this paper we have used all three siRNAs targeting H2A, H2B and H2C subunits independently, clearly demonstrating an increase in R-loop accumulation (Fig. 4e-f) and inflammation (Fig.5b and Supplementary Fig.7b), irrespective of which RNase H2 subunit was depleted. These data support our overall conclusions.

2. I would like to see a quantification of the western bands in Extended data Figure 1e to see how it compares to Figure 1a-b. Comparing the Extended Data Figure 1e with Figure 1a,

the fraction of chromatin bound RNase H2 subunits appears to be very similar after siLuc as after depletion of the different components of the RNase H2 complex. It seems that the depletion affects mostly the free form of RNase H2 but shows little effect on the chromatin-bound form, or that the fraction of bound form is very small.

*We provide the quantification of independent fractionation experiments (n=3-4) and more representative western blot in the **new Supplementary Fig. 1e**. Even though the efficiency of the RNase H2A depletion is lower in the chromatin fraction (~70%) compared to depletion of the free form of RNase H2A (~95%) (**Fig 1b**), we have observed a decrease of RNase H2A bound to the studied genes by ChIP-qPCR (**Fig. 1h, j and l**). This ensures that the reported DRIP changes for the analysed genes are likely to be direct effects of RNase H2 deficiency in the chromatin fraction.*

3. Because RNase H2 would be expected to associate with chromatin as part of its RER function, it is important to show convincingly that the association of RNase H2 and RNA Pol II is strong in quiescent fibroblasts, which the data presented in Extended Data Figure 3b does not appear to support well. The inclusion of clear western blot data is needed to help substantiate this association.

*We provide a new clearer WB in quiescent fibroblasts, supporting the association of RNase H2 with the total Pol II and transcription-proficient Pol II, phosphorylated at Ser2 and Ser5 residues of the CTD, independent of DNA replication (**new Supplementary Fig 3b**).*

4. In figure 4b it is shown that RNase H2 peak coincides with the DRIP peak in cells with wt RNase H2, suggesting that RNase H2 binds but doesn't cleave R-loops enriched regions. Could an explanation for this be proposed? Perhaps RNase H2 binds all or most R-loops but only processes a small portion of them.

Indeed, in Figure 4b we observe co-localization of DRIP and RNase H2 peaks, suggesting that RNase H2 is bound to many genes. R-loops occur co-transcriptionally and occupy ~5% of the human genome under physiological conditions (Sanz et al., 2016). They are over-represented at promoter and termination regions, which have higher Pol II occupancy. It is likely that through its interaction with Pol II, RNase H2 may be required to maintain physiological levels of R-loops, by removing occasional R-loop excess co-transcriptionally. This function may be particularly important for short genes. Long genes may have other mechanisms to deal with R-loops (e.g. helicases such as UAP56/DDX39B, Perez-Calero et al, 2020) to avoid the cleavage of the nascent transcript. We now clarify this model in the revised discussion (ms page 8).

Moreover, it has been suggested that RNase H (1 and 2) cleave R-loops formed in the absence of Top1 which impairs transcription elongation in rDNA and tRNA genes in yeast (El Hage et al., 2010; El Hage et al., 2014). We now discuss the possibility that RNase H2 may be required for efficient transcriptional elongation and may be involved in transcriptional elongation checkpoint (ms page 8). Lack of this checkpoint control, will result in accumulation of abortive transcripts forming R-loops in RNase H2-deficient cells.

5. That the depletion of RNase H2 induces immune-related mRNAs is something that has been described in several systems, and it was previously shown to be associated with its RER activity. To demonstrate an involvement of the R-loop processing activity of RNase H2, the authors used overexpression of RNase H1. They showed that increased RNase H1 activity decreased the expression of immune-related genes. However, overexpression of RNase H1 may have additional effects, such as decreasing mtDNA replication. In addition, it has not been established whether RNase H1 can functionally replace RNase H2 in processing R-loops in mammalian cells. A much better experiment would be to delete the RNASEH2A

gene and express the RNaseH2A-RED mutant, which has been successfully used for separating the two activities of RNase H2 and is specific for the removal of R-loops.

We agree that the RNase H2A RED mutant would be a great tool to study R-loop-associated function of RNase H2. However, it was recently shown that the RED mutant has a significantly decreased activity (<50% of the WT) on RNA/DNA hybrids both in vivo and in vitro likely due to reduced substrate affinity (Benitez-Guijarro M et al, 2018). Furthermore, RNase H2 is a heterotrimeric enzyme, and overexpression of the catalytic subunit alone is not sufficient to rescue its activity (Benitez-Guijarro M et al, 2018). Therefore, upon deletion of RNASEH2A, the expression of all three subunits is required for the rescue. Stable depletion of RNase H2 results in accumulation of ribonucleotides in the genomic DNA and genome instability (Zimmermann M, 2018; Lim YW, 2015; also see answer to Reviewer 3 point 4). Therefore, to avoid the problem of reduced activity of RED mutant and the use of RNASEH2A KO cells, we have over-expressed RNase H1 in HEK293T cells depleted for RNase H2A by RNAi. We now show that over-expressed RNase H1 can functionally replace RNase H2 in processing R-loops in the studied genomic loci by DRIP-qPCR (new Fig. 4h and Supplementary Fig. 6b), supporting the role of RNase H2 in R-loop resolution.

We also note that we can not fully exclude the possibility that RER and R-loop activities of RNase H2 are functionally linked, adding an additional layer of complexity. As we state in the discussion, the accumulation of rNMPs in the genomic DNA may also trigger replication stress, DNA damage and R-loop formation (ms page 9).

Overexpressed RNase H1 construct used in our experiments does not impact mitochondrial replication as it lacks mitochondrial localization signal (MLS) and hence is expressed only in the nucleus (Cerritelli et al., 2003). We have now added this information to the Materials and Methods section.

6. Does the RNase H1 overexpression system express only the nuclear form of the enzyme, or does it also express the mitochondrial form? This construct should be clearly described.

RNase H1 over-expression construct used in our experiments lacks mitochondrial localization signal (MLS) and hence is solely expressed in the nucleus (Cerritelli et al., 2003). We now provide this description in the Methods section.

Reviewer #2 (Remarks to the Author):

This manuscript describes the genome-wide profile of RNaseH2 binding to the human genome. The main findings are that RNaseH2 associated with transcribed regions of the genome in a transcription-dependent manner and through physical association with RNA polymerase II. Loss of RNaseH2 function leads to defects in transcription that appear selective to specific types of transcripts, namely short intronless transcripts. As expected, loss of RNaseH2 leads to increases in R-loops at some loci. These R-loops can lead to DNA damage caused by the XPF/XPG nucleases and activate transcription of inflammatory response genes, which is relevant to the RNaseH2-deficiency disorder Aicardi-Goutieres syndrome. Overall, the manuscript is clear and straightforward and presents a novel finding that RNaseH2 regulates transcription through association with RNA polymerase II and transcribed genes.

I have a number of technical questions, and suggestions for additional analyses that could improve the work. In short, while Figures 1-3 are novel, Figures 4 and 5 primarily serve to validate the literature but do not add much new insight. In the end this leads to a model which lacks some key insights; specifically how and when does RNaseH2 interact with RNA polymerases? how does rapid degradation of a co-transcriptional R-loop promote efficient transcription, and what does this have to do with short genes? How do these R-loops trigger an inflammatory response?

A deeper exploration of some of these questions, even using the authors existing data and some assays they already have working would dramatically strengthen the manuscript.

Response: *We thank this reviewer for her/his positive comments and finding our manuscript 'clear and straightforward' and presenting 'a novel finding that RNaseH2 regulates transcription through association with RNA polymerase II and transcribed genes'. We want to highlight that the role of RNase H2 in co-transcriptional R-loop resolution (Figures 1-4) and the role of R-loops in inflammatory response in RNase H2-deficient cell (Figure 5) in mammalian systems has not been reported so far and therefore both represent novel discoveries. Below we provide additional experiments and controls to support the role of RNase H2 in transcription, R-loop resolution and inflammatory response through DNA break production. These data are also incorporated in the manuscript to support our model and provide more clarity to our results.*

Major issues:

1. The EU staining data seems to also show that nucleolar intensity is decreased. The authors should test whether RNaseH2 also affects RNA Polymerase I and III transcription, and whether RNA polymerase I and III can be observed in pulldowns of RNaseH2 (as in Fig. 3A). This is a missed opportunity to learn more about how RNaseH2 is actually recruited. For example, if all three polymerases could recruit RNaseH2 then it would suggest that a shared subunit may be responsible for the interaction.

We have now provided the quantification of total, nucleoplasmic and nucleolar EU signal by using an automated analysis with high-content microscopy in both HeLa and HEK293T cells (>20,000 cells analysed per condition) (new Fig. 3c and Supplementary Fig. 3d-e). Indeed, we detected a decrease in EU staining both in the nucleoplasm and nucleolus of RNase H2-deficient cells suggesting a decrease in Pol II and Pol I transcription in these cells. This is further supported by our new results demonstrating that RNase H2 also interacts with RNA Pol I in co-IP experiments, suggesting its potential role in rRNA genes transcription and R-

loop biology (new Supplementary Fig. 3c). RNase H (1 and 2) plays a role in Pol I transcription in yeast (El Hage et al., 2010; El Hage et al., 2014). In mammalian cells, it has previously been reported that RNase H1 is involved in rDNA R-loop processing in nucleolus (Shen et al., 2017; Abraham et al., 2020). However, it is currently not clear if RNase H1 and RNase H2 have redundant roles in these processes.

We have not observed RNase H2 interacting with RNA Pol III suggesting that most probably RNase H2 is not required for the transcription of Pol III genes in human cells.

2. The correlation between R-loop levels and RNaseH2 occupancy is quite strange. Naively one would expect higher RNaseH2 at a locus to reduce the lifetime and amount of R-loops present. The authors suggest other factors may compensate. I wonder if catalytic mutants of RNaseH2A would behave the same way in terms of their effects on transcription. The authors could attempt to rescue their siRNA cells with WT or mutant RNaseH2A to determine if the R-loop resolution function is required for the observed effects on global transcription.

Our data show that RNase H2A occupancy mirrors RNA Pol II occupancy and RNase H2A co-IPs with RNA Pol II. We suggest that RNase H2 may be part of transcriptional complex ready to act on R-loops when their level exceeds a physiological threshold, especially on short genes. Therefore, in view of this model, we do not necessarily expect that higher RNase H2 occupancy would correlate with a reduced amount of R-loops.

*Unfortunately, the experiment to address the role of R-loop-associated function of RNase H2A in transcription is technically challenging because it requires expression of all three RNase H2 subunits together. Indeed, using CRISPR/Cas9 editing we have generated RNASEH2A KO HEK293T cells (**Response Fig R5a**). These cells demonstrated an increase in micronuclei accumulation, as previously reported, and an increase in R-loops on specific genomic loci by DRIP-qPCR, in line with our data in RNase H2A-deficient HeLa cells (**Response Fig R5b-c**). Over-expression of a plasmid encoding all three RNase H2 subunits provided by A.Jackson's lab, caused a reduction of elevated R-loops in RNASEH2A KO cells, supporting a role of RNase H2 in R-loop resolution (**Response Fig R5**). Unfortunately, these cells were not sufficiently healthy to generate reproducible transcriptional data, most probably due to cellular toxicity associated with the long-term depletion of RNase H2 and heavy transfection procedure. This prevented us from testing the role of catalytic mutant of RNase H2A in transcription. These experiments are only possible if we or others in the field are able to generate better cellular models of the disease suitable to study R-loop biology and gene expression. In the revised version of the manuscript, we now discuss that nuclease-independent functions of RNase H2 cannot be completely excluded (ms page 8). Protein-protein interactions with components of the transcriptional complex may be important for RNase H2 role in gene expression.*

3. The induction of immune related genes by qPCR Figure 5f seem to normalize both the siLuc and siXPG to 1 for the mRNA levels. I think the siXPG should not be normalized and should be compared directly to siLuc. Does XPG or XPF knockdown alone increase inflammatory gene expression? It may be that the relative increase in siRNaseH2A is lower, but if the background in siXPG is much higher it could change the interpretation. The way the data is presented does not allow for this kind of comparison.

We now present a figure where all data points are directly compared to siLuc (new Supplementary Fig. 8d and 8g). Indeed, individual depletion of XPG or XPF (to a lesser extent) leads to some increase in inflammatory gene expression. However, importantly, simultaneous co-depletion of XPG+RNase H2 or XPF+RNase H2 does not result in enhanced

immune response, compared to single depletions. This is in contrast to co-depletion of RNase H2 and R-loop resolving helicase AQR, resulting in an enhanced immune response (Supplementary Fig. 7d-e). This suggests that both XPG and XPF nucleases are required for this R-loop-driven inflammatory gene expression.

Both XPG and XPF are DNA repair enzymes involved in global nucleotide excision repair (GG-NER), transcription coupled repair (TCR) and other repair pathways, thus their deficiency may cause DNA damage and thereby inflammatory gene expression. Moreover, it has been shown that mice lacking ERCC1, the co-factor of XPF, have chronic inflammation (Karakasilioti et al., 2013). Therefore, we have used initial normalization for both siXPG/siXPF and siLuc cells to examine the contribution of these endonucleases to the immunity in RNase H2-deficient cells beyond their roles in other pathways (i.e. by comparing XPG-proficient (siLuc cells) and XPG-deficient (siXPG cells) cells upon depletion of RNase H2A).

4. In Figure 5e, Might the authors have expected that XPF depletion on its own would have caused DNA damage. The box plot should perhaps show the individual data points, it seems like the siXPF sample has a long tail of cells with damage. Is the distribution normal? Can the authors do a statistical test of siXPF versus control to see if there is an increase in comet tail moment? Currently the siXPF bar (purple) is not compared to the control siRNA bar (white) as far as I can see. This should be done and discussed. Which post-hoc statistical test was used should be indicated in the legend also. In addition, if the damage is really due to R-loops RNaseH1 expression could be included as an additional control, especially since XPF/XPG knockdown do not completely suppress the damage.

As described in the response to point 3, XPF is involved in DNA damage repair and its depletion may result in DNA damage as reported by both γ H2AX and phospho 53BP1 foci staining in non-replicating fibroblasts (Cristini et al. 2019; Fig S7C) and COMET assay in U2OS cells (Li et al. 2019; Fig.6C). We have now enlarged our alkaline COMET analysis in Fig 5e to include more cells ($n>400$) and provide individual data points (Supplementary Fig 8c). We have performed One-way ANOVA statistical test for multiple comparisons (the mean of each dataset is compared with the mean of every other dataset) and also Tukey test as a post-hoc statistical test, as indicated in the Figure legend. Based on this analysis, the increase in COMET tail moment in siXPF vs siLuc cell is not significant ($p=0.6352$). In the **Response Fig. R1** below we also provide the average of the alkaline COMET tail moment means of 3 independent biological experiments. These data show that despite an increase in COMET tail moment in siXPF cells compared to siLuc cells in agreement with the cited literature, this is not significant using one-way ANOVA, Tukey statistical test ($p=0.9777$), which takes into account all the groups in the dataset.

Moreover, as suggested by the reviewer, we now provide the alkaline COMET assay in HEK293T cell line inducible for RNase H1 showing that RNase H1 overexpression partially prevents DNA breaks accumulating upon RNase H2A depletion (**new Fig. 5d**). These results further support a contribution of R-loops to accumulation of DNA breaks in RNase H2-deficient cells.

5. Is it possible that the immune genes are directly affected by RNaseH2A knockdown due to the transcriptional role? To address any effects the authors could analyze the DRIP, RNAPolIII and RNaseH2A occupancy at all of the reporter genes (TNF, STING etc). This is important to rule out direct effects on transcription as opposed to DNA damage-cytoplasmic DNA induced activation. Similarly, the authors rely on mRNA expression but looking at protein level induction and possibly at micronuclei as a cause of STING activation would fill out the story. Additional data to help us understand how these R-loops trigger an inflammatory response would help better justify the focus on AGS in the title.

Below we now provide the RNase H2A/DRIP/chrRNA occupancy screen shots (Response Fig. R2). chrRNA-seq analysis of these genes shows their increased transcription. However, RNase H2 is not bound to these genes based on lack of RNase H2 IP enrichment over input in ChIP-seq lanes and therefore ruling out a direct effect of RNase H2 KD on transcription of immunity genes. Moreover, most of these genes are expressed at a low level, resulting in a very low DRIP signal which is close to the RNase H-digested background control DRIP sample.

We also provide new experiments showing a significant increase of micronuclei in RNase H2A-depleted HeLa cells (new Supplementary Fig 7c), which are in agreement with previous literature (Pizzi et al; 2015). Finally, we demonstrate that R-loops contribute to DNA breaks in RNase H2A-depleted cells by alkaline comet assay with RNase H1 overexpression (new Fig 5d). Taken together, these data suggest that persistent R-loops in cells deficient for RNase H2 promote DNA break formation which contributes to the inflammatory response.

Response Fig. R2: Overlay between ChrRNA-seq, RNase H2A ChIP-seq and DRIP-seq profiles for IFNGR1, PTGS2, OAS1, and ISG20 genes in HeLa cells. For ChrRNA-seq and DRIP-Seq HeLa cells transfected with siLuc or siRNASEH2A were used.

6. The bias toward short genes is not really explained because RNaseH2 and DRIP signal still occupy a lot of normal genes. If it is only when RNaseH2 is depleted that the effect on short genes is seen, then why does it RNaseH2 occupy all of those other genes. This has got to be explained better. See my comment above (#2) on testing whether the R-loop resolution activity is even required for the effects on transcription. The identity of genes where R-loops are lost in RNaseH2-knockdown would be another interesting place to reanalyze and get additional insights.

As requested, we have performed a Gene Ontology analysis of genes where R-loops are lost or gained in RNase H2-deficient cells. However, we have found no stand-out groupings (no group with a fold change >1.5) that might help rationalize why some genes are affected whereas others are not (**Response Fig. R3** above).

- Why Rnase H2 is required on long genes? Later on Page 7 the authors hint at a very exciting model for why there are redundant pathways at long genes. Reading between the lines are you saying if a short gene forms an R-loop and gets 'stuck' the cell can afford to just make it over again, so degradation by RNaseH2 is preferred. For very long genes this is a bad strategy because degradation of that long transcript wastes a lot of energy and time to remake the long transcript, so other pathways predominate? I'd like the authors to confirm and expand on this model if they agree. Could analysis of DRIP peak changes in other siRNA conditions confirm that RNaseH2-effects short genes while other pathways affect long genes?

- This paper (<https://pubmed.ncbi.nlm.nih.gov/32747416/>) generates DRIP data in a few other cell lines with other siRNAs, maybe the gene length effects could be analyzed in these data sets and compared with the gene length of DRIP-gain genes RNaseH2A?

a DRIP-seq: DDX5 KD in intron-containing genes

b DRIP-seq: PRMT5 KD in intron-containing genes

c DRIP-seq: XRN2 KD in intron-containing genes

d RNA-seq

Response Fig. R4: Meta-analysis of DRIP-seq on short (<2 kb) and long (>10 kb) intron-containing genes in U2OS cells depleted for DDX5 (a), PRMT5 (b) and XRN2 (c).

The same genes analyzed in siRNASEH2A experiments (Supplementary Fig. 5) have been analyzed using DRIP-seq datasets from Villarreal et al., 2020. Left panels represent short genes (<2 kb) and right panels correspond to long (>10kb) intron-containing genes. d) Boxplots of RNA-seq transcript signals from short and long intron-containing genes from U2OS cells transfected as in a-c. *** $p < 0.001$, **** $p < 0.0001$ (Friedman test).

We have now expanded the discussion section on the role of RNase H2 in co-transcriptional R-loop regulation of short vs long genes. RNase H2 may be present on long genes as a part of the transcriptional complex. Many R-loop regulators (e.g. SETX, DHX9, DDX23, DDX5, BRCA1, BRCA2 and XRN2) are known to be associated with the transcriptional complex. However, each regulator seems to have both overlapping and unique functions in R-loop resolution. Therefore, one possibility is that RNase H2 represents a backup mechanism to overcome an excess of R-loops on long genes when other redundant pathways have failed. We also discuss the possibility that long genes may accumulate R-loops at earlier time points after RNase H2 knockdown (before 48-72 h) compared to short genes (ms page 8). Therefore, the R-loop removal backup pathways may have already resolved them at the experimental time-points used in this ms.

*In addition, as suggested, we have carried out bioinformatics analysis of published DRIP-seq datasets in U2OS cells depleted for the helicase DDX5, the arginine methyltransferase PRMT5 or the exoribonuclease XRN2 (from Villarreal et al., 2020). All these factors have been implicated in R-loop metabolism in several independent publications (e.g. Mersaoui et al., 2019, for DDX5; Zhao et al, 2016, for PRMT5; Morales et al., 2016, for XRN2). Our new analysis shows that R-loops accumulate over both short (<2kb) and long (>10kb) genes upon depletion of DDX5, PRMT5 or XRN2, without major effects on their expression (apart from DDX5 KD which slightly increases their expression) (**Response Fig. R4a-d**). We now provide further discussion of this point in the revised manuscript (ms page 8), indicating that RNase H2 acts preferentially on short genes whereas TOP1 (Manzo et al., 2018) and UAP56/DDX39B (Perez-Calero et al., 2020) seem to mainly control R-loops over long genes. Other R-loop regulators, including DDX5, PRMT5 or XRN2, regulate R-loop level independently of the gene size.*

7. I do not really think the model is explained well. How does R-loop degradation lead to efficient transcription? It would seem that it would degrade the transcript, how does this help efficiency? The authors may need to consider literature on stalled RNA polymerase and associated R-loops. Could the stabilized R-loops reflect a backlog of stalled polymerase that needs to be cleared for productive transcription?

We now provide further clarification and discussion of our working model. We propose that RNase H2 is required to resolve co-transcriptional R-loops if they exceed the physiological level. Thus, through its interaction with Pol II, RNase H2 is involved in checkpoint control to promote efficient transcription. In RNase H2-deficient cells, R-loops accumulate and interfere with transcription likely through Pol II stalling, especially on short genes. This is reminiscent of the proposed model for RNase H (1 and 2) in transcription of rDNA genes in yeast (El Hage et al., 2010; El Hage et al., 2014). Indeed, Top1-deleted yeast strains lacking RNase H activity accumulate R-loops at rDNA genes, which cause impaired transcription elongation, Pol I pileups and reduced rates of pre-rRNA synthesis. Previous work in mammalian systems also suggested that R-loop accumulation promotes Pol II stalling in termination regions (Skourti-Stathaki et al, 2011) and defects in transcriptional elongation (Groh et al, 2014). We now further discuss these points in the discussion section of the revised paper (ms page 8).

Minor issues:

- Page 4. The concluding sentence of the section “Taken together, our results imply that RNase H2 plays a role in gene transcription” is overstated. At this point in the story, the authors just finished show that RNaseH2 is recruited to the genome in a transcription dependent manner, it does not imply that RNaseH2 has a role in transcription.

Thank you for pointing this out. To better reflect our experimental results, we have toned down the text to state that ‘our results suggest that RNase H2 may play a role in gene transcription’.

- In figure 1A, siRNaseH2A shows complete KD. In figure 5C siRNaseH2A is not as good. I recognize that the cell lines are different but the differences are quite striking. Could this low penetrance of knockdown in Figure 5 have affected the phenotype?

We observe that the efficiency of the RNase H2A RNAi-mediated knock-down is similar on mRNA levels in HEK293T and HeLa cells (e.g. compare Supplementary Fig 1c and 6a). However, on a protein level, it is lower in HEK293T (Fig.5c corresponding to new Fig. 4g; Supplementary Fig. 3d) compared to HeLa cells (Fig.1a-b; Supplementary Fig.1c-e). This may reflect the differential stability of the RNaseH2 complex in these two cellular models. However, despite these differences, transcriptional defects, R-loops, DNA damage and increased expression of inflammatory genes were consistently observed upon siRNaseH2A in both HeLa and HEK293T cell lines (e.g. compare Fig 5b and 5c; Fig.5d and 5e), further supporting our conclusions. Generally, we observed that a more efficient RNase H2A depletion leads to a stronger phenotype.

- Can you justify use of the fibroblast model versus a neuronal model system. A more relevant model system would strengthen the arguments. Indeed, the title of the paper really suggests a focus on AGS pathology, which is not well developed here

We used fibroblasts here as a model for non-replicative cells (and not AGS model) as it has been extensively used in the literature (e.g. Lin et al., 2008; Huang et al., 2010; Cristini et al. 2019). Experiments in fibroblasts confirmed the role of RNase H2 in transcription, which is independent of its RER function in DNA replication.

- Information about what constitutes a gained DRIP peak or a lost DRIP peak should be included in the main text on Page 5. A reader will see that the total number of peaks does not add up with the gained/lost peaks. I assume this is due to a threshold of changes in peak reads but it would help to clarify exactly what the authors consider a gained/lost peak.

The detailed description of peak quantification and assignment to the ‘gain’/‘loss’ categories is now provided in the Supplementary Methods ‘ChIP/DRIP-seq data processing’ section of the paper. In brief, total number of peaks for siLuc and siRNaseH2A samples were obtained by comparing each IP to its respective input, and normalized to IP+RNase H control to avoid unequal non-R-loop background level in DRIP-seq. The peaks were assigned into ‘gain/loss’ categories based on the ratio of the reads in siRNaseH2A vs siLuc samples (<0.5: decreased/lost; >2: increased/gain; others: unchanged).

The higher number of gained/lost peaks compared to the total number of DRIP peaks for siLuc and siRNaseH2A is due to the difference in the peak calling algorithm used (MACS2 callpeak vs MACS2 bdgdiff) and the breakdown of the initial peaks into smaller peaks when one or more parts of an initial peak are enriched differently in siLuc or siRNaseH2A samples. We feel that Methods section now provides sufficient information about peaks assignment. However if required, we can also provide it in the main text of the paper.

- The authors state in places that RNaseH2 forms a complex with RNA polymerase II. The only evidence shown is that they interact by co-IP and western. I think the word ‘complex’ may be overstating the results. We do not learn a lot in this study about when and how RNaseH2 interactions with RNA polymerase II occur.

We fully agree with this comment and therefore we have now toned down the text to state that ‘RNase H2 associates with the Pol II complex’ in the text. Since this paper is primarily focusing

on the role of RNase H2 in R-loop resolution, we feel that the specifics of RNase H2 and Pol II interaction are beyond the scope of this paper and require a separate investigation.

- The rationale for using an alkaline comet assay in Figure 5 is unclear. Are the authors expecting double strand breaks? If not, then how could ssDNA breaks trigger an inflammatory response? A neutral comet assay might help them better connect the damage to STING activation.

*To assess the overall level of DNA damage, alkaline COMET assay was used in Fig.5, which detects both SSBs and DSBs. Both ssDNA and/or RNA/DNA hybrids released from R-loops by combined ssDNA cleavage events mediated by XPG/XPF (as reported in Cristini et al. 2019) could also elicit an inflammatory response (Shen et al, 2015; Coquel et al., 2018). Following the reviewer's suggestion, we have also carried out Neutral comet assay (new **Supplementary Fig. 8a**). We observed an increase in DSBs in RNase H2-depleted cells, which is in line with previous studies. This suggests that DNA damage drives STING activation in AGS conditions.*

- The legend for Figure 2C says ActD is compared to a DMSO vehicle but the figure says EtOH. This should be clarified.

We have changed the legend accordingly.

- The legend for Figure 2 also refers to panel g, as (d).

We have changed the legend accordingly.

- The legend for Figure 1 labels intronless JUNB as k, while it should be l.

We have changed the legend accordingly.

Reviewer #3 (Remarks to the Author):

This is a nice paper showing a kind of unexpected role of RNASEH2 during transcription by RNA polII. Using DRIP-seq and ChIP-seq to analyze DNA-RNA hybrids and the presence RNA polII and RNASEH2A, the authors show an enrichment of RNASEH2A at the 5' end region of genes. Removal of RNASEH2 leads to an increase of hybrids mainly at the regions where RNASEH2A is enriched in normal cells that, as expected from previous studies, is associated with DNA damage. At least part of this damage is produced by structure-specific endonucleases, which is one mechanism of break formation. Consistent with the current literature, damage leads to an enhancement in the RNA levels of genes of the immune response. In general, the manuscript provides new and interesting results that make it a candidate for Nat Comm. I have few comments added below to improve the manuscript, but also some important requests. It seems that the manuscript provides ChIP-seq and DRIP-seq data performed only once. At least the DRIP-seq and ChIPseq of RNH2A should be performed minimally twice to show that results are repetitive, to make this a Nat Comm article. In addition the model proposed has to be better justified. It is pretty counterintuitive and the data are open to additional interpretations.

We would like to thank this referee for the positive response and highlighting our 'new and interesting results' on RNase H2 function in transcription and co-transcriptional R-loop resolution. Starting from the original submission of this paper both DRIP-seq and RNase H2A ChIP-seq experiments presented were performed twice as also indicated in the figure legends and methods section (all the raw files are deposited in GEO (GSE146970; password for reviewers is: exmjgaowhbavrif). To further verify these data, we also provided extensive DRIP-qPCR validation for individual genomic loci throughout this paper. We have now also experimentally addressed multiple comments, as requested, and provide a fuller and hopefully better description and justification to the model in the discussion section of the paper.

Specific comments

1. It would be good to justify why they chose for DRIP-qPCR of related genes RNU1 and RNU2

Both U1 (RNU1) and U2 (RNU2) snRNA genes are examples of snRNA gene category. These are separate genes but they are unrelated, apart from belonging to the same snRNA gene category (in addition to others such as U4, U5, U6 etc snRNA genes).

2. Rather than showing the metaplot of intron-plus versus intron-less genes, which really reflects a comparison of short versus long genes, it would be more informative to see the data of ChIP-seq of exons versus introns.

We have not detected any specific enrichment of RNase H2 or R-loops in exons vs introns, hence we have not presented these data in such a way. However, we did observe a clear statistical trend of RNase H2 requirement for R-loop resolution in short vs long intron-containing genes, as shown in the paper.

3. Authors cannot claim that RNASEH2 interacts with RNAPolII as in page 5: “our results reveal that RNASEH2 is a part of the PolII complex which acts to promote transcription”. For such an affirmation authors would need to show the purification by affinity chromatography and to identify the proteins by MALDI-TOFF, and not just by Westerns of co-IPs. The use of benzonase is not sufficient to conclude that RNASEH2 associates with PolII in a stable manner regardless of DNA with the results provided. The data indicates that RNASEH2 associates with RNA polII, but not that it forms a complex with it.

We fully agree with this comment and therefore we have now toned down the text stating that 'RNase H2 associated with the Pol II complex' in the text.

4. The conclusion that RNASEH2 resolves co-transcriptional R loops needs to be further substantiated. Authors should produce a cell line expressing a catalytically-dead RNASEH2 to show that the nuclease activity is required for R loop resolution. This can be done by RT-qPCR in several of the genes detected to accumulate R loops in the DRIPseq. Alternatively, authors might consider the possibility of reducing R loops by overexpressing the three subunits of RNH2 together

The experiments to assess the role of nuclease function of RNase H2A in R-loop resolution are technically challenging because they require expression of all three RNase H2 subunits together. Indeed, using CRISPR/Cas9 editing we have generated RNASEH2A KO HEK293T cells. These KO cells demonstrated a loss of RNase H2A expression which correlated with a reduction of RNase H2C expression and an increase in micronuclei accumulation, as previously reported (Mackenzie KJ et al, 2017) (Response FigR5a-b). In these KO cells we have observed an increase in R-loops on specific genomic loci by DRIP-qPCR (Response Fig R5c), in line with our data in RNase H2A-deficient HeLa cells (Figure 4f). Though this increase in R-loop levels was less pronounced at later cellular passages, likely due to appearance of compensatory mechanisms interfering with the transcriptional program and R-loop homeostasis, making these cell lines suboptimal models to study the role of RNase H2 in R-loop homeostasis.

Nevertheless, as suggested by the reviewer, we over-expressed a plasmid encoding all three RNase H2 subunits, provided by A.Jackson's lab. This caused a reduction of elevated R-loops in RNASEH2A KO cells, supporting a role of RNase H2 in R-loop resolution (Response Fig R5d-e). We prefer not to show these data in the ms due to limitations of these RNase H2A KO cellular models for transcription/R-loop biology studies as discussed above. We have additionally demonstrated that over-expression of RNase H1 results in a decrease of R-loop levels elevated by RNase H2 depletion further supporting the role of RNase H2 in co-transcriptional R-loop resolution (new Figure 4h).

Response Fig. R5: RNase H2 is required for R-loop resolution (a) Western blot of RNase H2A and H2C in HEK293T WCE of parental, CRISPR/Cas9 control and RNASEH2A-KO cell lines. (b) Micronuclei analysis of parental WT and RNASEH2A-KO cell lines (means \pm SEM; n=2). (c) DRIP-qPCR analysis on the HIST1H1E gene in RNASEH2A-KO cell lines. Values are normalized to TES amplicon in the parental cell line (means \pm SEM; n=3), * p <0.05, ** p <0.01, **** p <0.0001 (two-tailed unpaired t-test). Gene diagrams are shown on the bottom panel. (d) Western blot analysis of parental WT or RNASEH2A KO clone transfected with eGFP plasmid or a plasmid encoding all three RNase H2 subunits together. RNase H2B subunit is tagged with eGFP. (e) DRIP-qPCR analysis of parental WT or RNASEH2A KO clone complemented with either eGFP or RNaseH2A/B/C plasmid. DRIP-qPCR values are normalized to TES amplicon of the HIST1H1E gene in the parental cell line (means \pm SEM; n=4), * p <0.05, *** p <0.001 (two-tailed unpaired t-test).

5. Authors should test whether RNASEH2 is recruited at regions accumulating R loops. Check at some of the RNH2 peaks whether this is reduced by RNASEH1 overexpression. This can be done in several regions by ChIP-qPCR. Wouldn't this be expected if RNASEH2 resolves those hybrids?

Using ChIP-seq and ChIP-qPCR we show that RNase H2 is recruited to the gene regions where R-loops accumulate in its absence (e.g. HIST1H1E, JUNB, RNU1, Fig 1). Our new results show that RNase H1 over-expression does not affect RNase H2 recruitment to the R-loop-containing loci (new Supplementary Fig 6c-d), most probably due to recruitment of RNase H2 through its interaction with Pol II, while RNase H1 is likely to be recruited to R-loops directly. Furthermore, literature in yeast suggests that RNase H1 and RNase H2 act on different R-loop subsets and function under different cellular circumstances (Chon et al, 2013; Zimmer et al, 2016; Lockhart et al, 2019). This suggests that RNase H1 is not likely to resolve R-loops at RNase H2-regulated loci in the presence of RNase H2.

6. I find the last point poorly connected with the rest of the ms. Certainly, the inflammatory response is a highly important phenomena that merits study, but authors should rationalize better how this relates with the rest of the paper. If the authors want to correlate the R loops with DNA damage as a way to strengthen conclusions is OK, but the rest seems to confirm that DNA damage will activate the inflammatory response. This needs to be discussed better. A main question is whether the action of RNASEH2 could be linked to a role in mitochondria or cytoplasm or other events. Can this be excluded?

We apologise for not making the last point clearer. Our results suggest that persistent R-loops contribute to DNA breaks which promote the inflammatory response in RNase H2-deficient cells. The contribution of R-loops to inflammatory response associated with RNase H2 depletion/mutations has not been reported in the literature so far. We propose that mis-regulated R-loops can drive DNA breaks by XPG/XPF nuclease-mediated cleavage, contributing to inflammatory response. We have now adjusted the results and discussion of the revised paper accordingly to better reflect this (ms page 7, 9).

Eukaryotic RNase H2 is a heterotrimeric complex with all three subunits being required for the catalytic activity (Jeong et al 2004; Chon et al 2009). RNase H2 complex is assembled in the cytosol and then it is imported into the nucleus in an RNase H2B-dependent manner (Kind et al., 2014), suggesting that the active complex mainly functions in the nucleus. Studies in yeast further suggest that RNase H2 is not involved in mitochondrial functions. Indeed, R-loops accumulate on mtDNA transcription units in the absence of RNase H1, but not in cells lacking RNase H2 (El Hage et al. 2014). Moreover, ribonucleotides incorporated by the yeast mitochondrial DNA polymerases are not repaired (Wanrooij et al 2017), further excluding a role of RNase H2 in mitochondria.

7. Fig. 4c with the plot of the DRIP-seq of snRNA data shows a peak downstream the TES. Authors need to discuss this result. Is this supposed to be a non-transcribed DNA region? What is the explanation for a hybrid signal higher than that observed inside the genes?

*Our Pol II ChIP experiments (see below **Response Fig. R6**) demonstrate the presence of Pol II downstream of snRNA TES, associated with snRNA genes transcription. This is in line with previously published data using nuclear run-on analysis (O'Reilly D et al. 2014). Our DRIP experiments suggest that this nascent RNA is prone to form higher level of R-loops compared to the gene body region. Previously we have published that R-loops downstream of poly A signal of mRNA genes are involved in promoting transcriptional termination of RNA pol II (Skourti-Stathaki et al 2011). Therefore, we predict that these R-loops at the 3' end of snRNA genes may be also involved in transcriptional termination.*

8. Fig. 4f. Experiments should show that RNH1 removes the DRIP-qPCR signals in the genes tested.

*We have now demonstrated that over-expression of RNase H1 in HEK293T cells results in a decrease of R-loop levels elevated by RNase H2 depletion, further supporting the role of RNase H2 in co-transcriptional R-loop resolution (new **Fig.4h** and **Supplementary Fig.6b**).*

9. Fig. 5. Authors need to show the absolute values of mRNA levels, so that the reader can compare different genes. It is easy to double the RNA levels of a gene that is low expressed versus a gene that is highly expressed. This is important considering that results shown correspond to RT-qPCR values and not complete mRNAs. It is critical that the levels of expression of the genes used as controls are similar to those of the immunity response.

The mRNA results presented in Fig 5 are based on absolute values of mRNAs, calculated using standard curve with cDNA dilutions in qPCR reactions. The data were normalised to siLuc sample to compare the immune induction for different genes in RNase H2-deficient cells.

*To compare the expression of different immunity and control genes we now also provide $\Delta\Delta Ct$ mRNA analysis in the **Response Fig. R7** below, which is based on Ct (threshold cycle). We have initially chosen two control genes TUBG2 (low expression comparable to most immune genes) and SELEBP1 (higher expressed) to accommodate for the potential differences in gene expression. Both ways of quantification show an increase in mRNA expression for immune genes, compared to control genes, in RNase H2-deficient cells. This is in agreement with the chrRNA-seq data in siLuc and siRNase H2 cells (Fig 5a). We can provide $\Delta\Delta Ct$ mRNA analysis in the Supplementary section, if you feel it is necessary.*

Response Fig. R7: Comparison of mRNA levels between immunity-related and control genes. RT-qPCR $\Delta\Delta\text{Ct}$ analysis of mRNA levels for all immunity-related genes and two control genes (TUBG2 and SELENBP1, indicated as SBP). For comparison, PTGS2 mRNA in siLuc condition is set to 1.

10. Authors should provide a main figure with a large region covering several genes, not just the length of one gene, so that we can compare the ChIP-seq, DRIP-seq, RNASEH2 pChIP-seq at once.

We have now included a screenshot covering multiple genes and showing the overlap between Pol II, RNaseH2 and R-loop distribution in new Supplementary Fig 4b-c.

11. Fig. 5 Considering that results shown correspond to RT-qPCR values, it would be relevant to show a scheme showing which regions has been RT-qPCR to make conclusions.

We now have included a scheme showing positions of RT-qPCR primers in Supplementary Fig. 7a.

12. Fig. 5e. Authors need to show that RNH1 suppresses the comet result, otherwise the result can be explained by the DNA-inserted ribonucleotide removal activity of RNASEH2

As requested, we have carried out alkaline comet assay in HEK293T cells over-expressing RNase H1 (new Fig. 5d). These new results show that RNase H1 decreases DNA breaks accumulating in the absence of RNase H2. This indicates that the DNA damage induced by the depletion of RNase H2 is at least partially R-loop-dependent.

13. I would reconduct the discussion on intron-less and intron-containing genes. The data show that short genes have DRIP-seq signal covering a large part of the gene, whereas intron-containing genes the signal is concentrated on the 5' end. It seems that R loops are accumulated at the 5' end of genes therefore. In long genes, R loops are not seen at the middle and 3' end of genes. With the data observed, it is difficult to think that RNASEH2 travels with the transcription machinery resolving R loops, since almost no effect or presence of RNASEH2 is observed in the second half of genes longer than 2-kb. It seems its activity is limited only to the 5' end of genes. Intron-less genes and intron-containing genes <2kb shows similar profile indeed. This may be important to discuss in the model, since it may well be possible that RNASEH2 has a role aborting the elongation of suboptimal transcripts that form R loops; otherwise it would be counterproductive for the cell to degrade any long nascent RNA forming short hybrids. Otherwise, how can authors propose that an active RNH2 that would be removing nascent mRNAs forming short stretches of hybrids leads to efficient transcription. Unfortunately authors do not know

directionality of the RNA forming the hybrids, since I am not sure whether RNASEH2 could remove hybrids formed with antisense RNA.

The DRIP-seq results confirm that the long genes have R-loops enriched over input within the body of the genes, however this signal is dramatically enriched over the 5' end of the gene (Supplementary Fig. 5f). In contrast, the level of R-loops is consistently high over the whole body of the short genes (Supplementary Fig. 5b and d). DRIP-seq profiles mimic the profiles of RNA pol II distribution over short and long genes, suggesting that RNase H2 may travel with Pol II during transcription. Indeed, it may be possible that RNase H2 is involved in aborting the transcription elongation of suboptimal transcripts resulting in R-loop accumulation and transcriptional decrease in cells deficient for RNase H2. This is reminiscent to the proposed model for RNase H (1 and 2) in transcription of rDNA and tRNA genes in yeast (El Hage et al. 2010; El Hage et al. 2014). Indeed, top1-deleted yeast strains lacking RNase H activity accumulate R-loops at rDNA and tRNA gene, which cause impaired transcription elongation, Pol I pileups and reduced rates of pre-rRNA and pre-tRNA synthesis. We now discuss this possibility in the discussion section of the revised paper (ms page 8).

In yeast, it has been suggested that RNase H (1 and 2) may cleave ncRNA in the R-loops formed over the rDNA ISG regions in strains lacking top1 (El Hage et al. 2010). However, it is not clear which RNase H is responsible for this function. Even though we cannot fully exclude a possibility that RNase H2 removes hybrids formed with antisense RNA, our data show that RNase H2 binding correlates with the level of Pol II sense transcription on specific genes (Fig 2a). Therefore, we predict that RNase H2 function is more likely to be associated with the sense transcription and therefore sense hybrid removal.

14. Authors should be homogeneous with the plots. Why for instance Ext Fig. 4c, e shows 0.5 kb and +1,5kb region upstream and downstream of genes, whereas g, I show -2 and +2,5 kb. Use in this and all other figures the same length of regions surrounding the genes, so that we can compare all data visually. The same criticism for fig. 1 and 3 metaplots. Indeed, it is not clear to me why the length upstream and downstream is not the same, making thus plots symmetrical.

For all genome-wide experiments (ChIP, DRIP and chroRNA), we have adjusted the window size of the presented plots to the gene size, as we now indicate in all corresponding figure legends. Therefore we now show all short genes (snRNA, intronless and histone) with a window size of -0.5/+1.5 kb and all long genes with the window size of -2/+2.5 kb. The shorter flanking sequences for the short genes are also required to eliminate the signal originating from the closely located genes (e.g. closely spaced histone genes). We analyzed longer length downstream of each gene (i.e. plots are not symmetrical) because we wanted to capture transcriptional signature downstream of pA signal (TES) prior to Pol II termination, which can extend for a couple of kb downstream of TES in human cells.

References:

1. Abraham, K et al. Nucleolar RNA polymerase II drives ribosome biogenesis. *Nature* Sep;585(7824):298-302 (2020)
2. Benitez-Guijarro, M. et al. RNase H2, mutated in Aicardi-Goutieres syndrome, promotes LINE-1 retrotransposition. *EMBO J* **37**(2018).
3. Cerritelli, S.M. et al. Failure to produce mitochondrial DNA results in embryonic lethality in Rnaseh1 null mice. *Mol Cell* **11**, 807-15 (2003).
4. Chon, H. et al. Contributions of the two accessory subunits, RNASEH2B and RNASEH2C, to the activity and properties of the human RNase H2 complex. *Nucleic Acids Res* **37**, 96-110 (2009).
5. Chon, H. et al. RNase H2 roles in genome integrity revealed by unlinking its activities. *Nucleic Acids Res* **41**, 3130-43 (2013).
6. Coquel F et al SAMHD1 acts at stalled replication forks to prevent interferon induction. *Nature*. May;557(7703):57-61 (2018).
7. Cristini, A. et al. Dual Processing of R-Loops and Topoisomerase I Induces Transcription-Dependent DNA Double-Strand Breaks. *Cell Rep* **28**, 3167-3181 e6 (2019).
8. El Hage, A., French, S.L., Beyer, A.L. & Tollervey, D. Loss of Topoisomerase I leads to R-loop-mediated transcriptional blocks during ribosomal RNA synthesis. *Genes Dev* **24**, 1546-58 (2010).
9. El Hage, A., Webb, S., Kerr, A. & Tollervey, D. Genome-wide distribution of RNA-DNA hybrids identifies RNase H targets in tRNA genes, retrotransposons and mitochondria. *PLoS Genet* **10**, e1004716 (2014).
10. Groh M, Lufino MM, Wade-Martins R, Gromak N. R-loops associated with triplet repeat expansions promote gene silencing in Friedreich ataxia and fragile X syndrome. *PLoS Genet*. May 1;10(5):e1004318 (2014).
11. Huang, TH., Chen, HC., Chou, SM., Yang, YC, Li, TK. Cellular processing determinants for the activation of damage signals in response to topoisomerase I-linked DNA breakage *Cell Res* Sep;20(9):1060-75 (2010).
12. Jeong, H.S., Backlund, P.S., Chen, H.C., Karavanov, A.A. & Crouch, R.J. RNase H2 of *Saccharomyces cerevisiae* is a complex of three proteins. *Nucleic Acids Res* **32**, 407-14 (2004).
13. Karakasilioti, I et al. DNA damage triggers a chronic autoinflammatory response, leading to fat depletion in NER progeria. *Cell Metab* Sep 3;18(3):403-15 (2013).
14. Kind, B. et al. Altered spatio-temporal dynamics of RNase H2 complex assembly at replication and repair sites in Aicardi-Goutieres syndrome. *Hum Mol Genet* **23**, 5950-60 (2014).
15. Li S, Lu H, Wang Z, Hu Q, Wang H, Xiang R, Chiba T, Wu X. ERCC1/XPF Is Important for Repair of DNA Double-Strand Breaks Containing Secondary Structures. *iScience* Jun 28;16:63-78 (2019).
16. Lim, Y.W., Sanz, L.A., Xu, X., Hartono, S.R. & Chedin, F. Genome-wide DNA hypomethylation and RNA:DNA hybrid accumulation in Aicardi-Goutieres syndrome. *Elife* **4**(2015).
17. Lin, CP, Ban, Y, Lyu, YL, Desai, SD, Liu, LF. A ubiquitin-proteasome pathway for the repair of topoisomerase I-DNA covalent complexes. *J Biol Chem*. Jul 25;283(30):21074-83 (2008).
18. Lockhart, A. et al. RNase H1 and H2 Are Differentially Regulated to Process RNA-DNA Hybrids. *Cell Rep* **29**, 2890-2900 e5 (2019).
19. Mackenzie, K.J. et al. cGAS surveillance of micronuclei links genome instability to innate immunity. *Nature* **548**, 461-465 (2017).
20. Manzo, S.G. et al. DNA Topoisomerase I differentially modulates R-loops across the human genome. *Genome Biol* **19**, 100 (2018).
21. Mersaoui SY et al. Arginine methylation of the DDX5 helicase RGG/RG motif by PRMT5 regulates resolution of RNA:DNA. *EMBO J*. Aug 1;38(15):e100986 (2019)
22. Morales, JC, Richard P, Patidar PL, Motea EA, Dang, TT, Manley JL, Boothman DA. XRN2 Links Transcription Termination to DNA Damage and Replication Stress. *PLoS Genet* Jul 20;12(7):e1006107 (2016).
23. O'Reilly, D, Kuznetsova OV, Laitem C, Zaborowska J, Dienstbier M, Murphy S. Human snRNA genes use polyadenylation factors to promote efficient transcription termination. *Nucleic Acids Res* Jan;42(1):264-75 (2014).
24. Perez-Calero, C. et al. UAP56/DDX39B is a major cotranscriptional RNA-DNA helicase that unwinds harmful R loops genome-wide. *Genes Dev* **34**, 898-912 (2020).
25. Pizzi, S. et al. Reduction of hRNase H2 activity in Aicardi-Goutieres syndrome cells leads to replication stress and genome instability. *Hum Mol Genet* **24**, 649-58 (2015).
26. Sanz, L.A. et al. Prevalent, Dynamic, and Conserved R-Loop Structures Associate with Specific Epigenomic Signatures in Mammals. *Mol Cell* **63**, 167-78 (2016).

27. Shen W, Sun H, De Hoyos CL, Bailey JK, Liang, XH, Crooke, ST. Dynamic nucleoplasmic and nucleolar localization of mammalian RNase H1 in response to RNAP I transcriptional R-loops. *Nucleic Acids Res.* Oct 13;45(18):10672-10692 (2017).
28. Shen YJ, Le Bert N, Chitre AA, Koo CX, Nga XH, Ho SS, Khatoor M, Tan NY, Ishii KJ, Gasser S. Genome-derived cytosolic DNA mediates type I interferon-dependent rejection of B cell lymphoma cells. *Cell Rep.* Apr 21;11(3):460-73 (2015)
29. Skourti-Stathaki, K., Proudfoot, N.J. & Gromak, N. Human senataxin resolves RNA/DNA hybrids formed at transcriptional pause sites to promote Xrn2-dependent termination. *Mol Cell* **42**, 794-805 (2011).
30. Wanrooij, PH et al. Ribonucleotides incorporated by the yeast mitochondrial DNA polymerase are not repaired. *PNAS* Nov 21;114(47):12466-12471 (2017).
31. Villarreal, O.D., Mersaoui, S.Y., Yu, Z., Masson, J.Y. & Richard, S. Genome-wide R-loop analysis defines unique roles for DDX5, XRN2, and PRMT5 in DNA/RNA hybrid resolution. *Life Sci Alliance* **3** (2020).
32. Zhao, DY et al. SMN and symmetric arginine dimethylation of RNA polymerase II C-terminal domain control termination. *Nature.* Jan 7;529(7584):48-53 (2016).
33. Zimmer, A.D. & Koshland, D. Differential roles of the RNases H in preventing chromosome instability. *Proc Natl Acad Sci U S A* **113**, 12220-12225 (2016).
34. Zimmermann, M. et al. CRISPR screens identify genomic ribonucleotides as a source of PARP-trapping lesions. *Nature* **559**, 285-289 (2018).

REVIEWERS' COMMENTS

Reviewer #1 (Remarks to the Author):

The authors of the revised manuscript entitled "RNase H2, mutated in Aicardi-Goutières syndrome, resolves co-transcriptional R-loops to prevent DNA breaks and inflammation" have addressed appropriately all my concerns. They have substantially improved the manuscript, providing new insights and further supporting their main conclusions. I have no reservations about the quality and novelty of the work.

Reviewer #2 (Remarks to the Author):

This was a revised resubmission of the article by Cristini et al. The authors have provided a thorough and thoughtful response to the requested changes, adding new data and descriptions as requested. The revised manuscript is much improved and further highlights the novelty of the results. I have no further concerns.

Reviewer #3 (Remarks to the Author):

The authors did a nice work with the revision. The manuscript is much improved. I am satisfied with the result.

Response to the Reviewers' Comments

We would like to thank all three reviewers for their positive comments and highlighting the novelty of our results and the substantial revisions, which we carried out. All three reviewers were satisfied with the quality of the manuscript and did not request any further changes to the paper.

Reviewer #1 (Remarks to the Author):

The authors of the revised manuscript entitled "RNase H2, mutated in Aicardi-Goutières syndrome, resolves co-transcriptional R-loops to prevent DNA breaks and inflammation" have addressed appropriately all my concerns. They have substantially improved the manuscript, providing new insights and further supporting their main conclusions. I have no reservations about the quality and novelty of the work.

Reviewer #2 (Remarks to the Author):

This was a revised resubmission of the article by Cristini et al. The authors have provided a thorough and thoughtful response to the requested changes, adding new data and descriptions as requested. The revised manuscript is much improved and further highlights the novelty of the results. I have no further concerns.

Reviewer #3 (Remarks to the Author):

The authors did a nice work with the revision. The manuscript is much improved. I am satisfied with the result.